# Shadow imaging for panoptical visualization of brain tissue in vivo

Yulia Dembitskaya[1,8], Andrew K. J. Boyce[1,2,3,8], Agata Idziak [1,8], Atefeh Pourkhalili Langeroudi[1,8], Misa Arizono[1,4], Jordan Girard[1], Guillaume Le Bourdellès[1], Mathieu Ducros [5], Marie Sato-Fitoussi[1], Amaia Ochoa de Amezaga[1], Kristell Oizel[6], Stephane Bancelin[1], Luc Mercier[1], Thomas Pfeiffer [1], Roger J. Thompson[2,3], Sun Kwang Kim [1,7], Andreas Bikfalvi [6] & U. Valentin Nägerl [1] ✉

Progress in neuroscience research hinges on technical advances in visualizing living brain tissue with high fidelity and facility. Current neuroanatomical imaging approaches either require tissue fixation (electron microscopy), do not have cellular resolution (magnetic resonance imaging) or only give a fragmented view (fluorescence microscopy). Here, we show how regular light microscopy together with fluorescence labeling of the interstitial fluid in the extracellular space provide comprehensive optical access in real-time to the anatomical complexity and dynamics of living brain tissue at submicron scale. Using several common fluorescence microscopy modalities (confocal, light-sheet and 2-photon microscopy) in mouse organotypic and acute brain slices and the intact mouse brain in vivo, we demonstrate the value of this straightforward 'shadow imaging' approach by revealing neurons, microglia, tumor cells and blood capillaries together with their complete anatomical tissue contexts. In addition, we provide quantifications of perivascular spaces and the volume fraction of the extracellular space of brain tissue in vivo.

The human brain is a structural engineering marvel where hundreds of thousands of miles worth of tightly packed axons and dendrites connect billions of neurons into a gigantic electrochemical network that generates memory, thought, and action. Its main animal model, the mouse brain, is about three orders of magnitude smaller, but its anatomical structure is similarly dense and complex[1].

Fluorescence microscopy is the method of choice for multiscale imaging of living brain tissue with high resolution. However, it typically relies on labeling sparse sets of brain cells, providing a fragmented and partial view of tissue anatomy. Electron microscopy and magnetic

resonance imaging are practically label-free and unbiased approaches, yet they either offer high spatial resolution or non-invasive live imaging, respectively, but not both together. Moreover, it remains challenging to combine them with other powerful neuro-technologies, such as $Ca^{2+}$ imaging, electrophysiology, and optogenetics because of big differences in hardware and spatial scales (but see recent efforts[2,3]).

Getting a more detailed and broader view of brain tissue is not only important for mapping the functional connectivity of neuronal circuits[4], it can also unearth useful information on anatomical context and tissue viability to assist in experiments and their interpretation.

[1]Interdisciplinary Institute for Neuroscience, CNRS UMR 5297 and University of Bordeaux, F-33000 Bordeaux, France. [2]Department of Cell Biology and Anatomy, Cumming School of Medicine, University of Calgary, Calgary, Alberta, Canada. [3]Hotchkiss Brain Institute, University of Calgary, Calgary, Alberta, Canada. [4]Department of Pharmacology, Kyoto University Graduate School of Medicine/The Hakubi Center for Advanced Research, Kyoto University, Kyoto, Japan. [5]Université de Bordeaux, CNRS, INSERM, Bordeaux Imaging Center (BIC), UAR 3420, US 4, F-33000 Bordeaux, France. [6]Université de Bordeaux, INSERM, Bordeaux Institute of Oncology (BRIC), U1312, Bat B2, Allée Geoffroy St Hilaire, 33615 Pessac, France. [7]Department of Physiology, College of Korean Medicine, Kyung Hee University, Seoul 02447, Korea. [8]These authors contributed equally: Yulia Dembitskaya, Andrew K. J. Boyce, Agata Idziak, Atefeh Pourkhalili Langeroudi. ✉e-mail: valentin.nagerl@u-bordeaux.fr

Breaking this impasse, super-resolution shadow imaging (SUSHI) introduced a new paradigm to visualize the anatomy and extracellular space (ECS) of living brain tissue with nanometric spatial resolution[5]. It is based on fluorescence labeling of the interstitial fluid and super-resolution stimulated emission depletion (STED) microscopy[6,7], casting all membrane-enclosed cellular structures as sharply contoured 'shadows' against a bright background of extracellular fluorescence.

Imaging with inverted contrast is immune to bleaching because the interstitial fluid provides an inexhaustible reservoir of fresh dye molecules. It is potentially also much less invasive and harmful because the fluorophore does not need to be introduced into the cells and any phototoxic by-products do not accumulate inside of them, but can dissipate in the ECS. The most minute cellular and extracellular compartments (e.g., axon shafts, peri-synaptic astrocytic processes, spine necks, synaptic clefts) can be discerned in super-resolved shadow images[5,8,9].

A recent study using the SUSHI approach and machine learning generated precise 3D reconstructions of brain tissue microstructures, even though the spatial resolution was limited to 140 nm[10], suggesting that the neuroanatomical ground truth can be established by light microscopy when augmented by advanced computational image analysis.

Hampered by high technical demands and limited availability of super-resolution technology, SUSHI has not yet been widely adopted. Moreover, the technique has been used mostly in organotypic brain slices[8,9,11], which offer optimal imaging conditions, and not yet in acute brain slices or intact brains in vivo, where sufficient labeling and image contrast for shadow imaging are harder to come by.

The aim of this study was to mitigate these difficulties and make shadow imaging more versatile, outlining a straightforward (and adoptable) method for capturing more fully the complexity and dynamics of the anatomical structure of living brain tissue even without super-resolution microscopy.

To this end, we have worked out solutions for achieving innocuous, high-contrast fluorescence labeling of the interstitial fluid in acute brain slices and the intact brain in vivo. We show that the inverted signal can be read out by regular microscopy techniques, such as confocal, 2-photon, and light sheet microscopy, providing fine-grained yet expansive views of the anatomical scenery in these major experimental preparations. We then demonstrate the utility of the approach to extract new and practical neuroanatomical information concerning microglia, neurons, tumor cells, and brain vasculature.

## Results and discussion

We started out by imaging organotypic hippocampal brain slices using an inverted confocal microscope (Fig. 1a). We chose the more popular Muller slices[12], rather than Gähwiler[13], for which the SUSHI technique had originally been developed. A Muller slice is grown on a light-scattering membrane support, but by turning it upside down and placing a metal ring on top, it is possible to get a direct and stable view from below.

We used a 93X glycerol-immersion objective (NA 1.3) equipped with a motorized correction collar to reduce spherical aberrations caused by the mismatch in refractive index between medium ($n \sim 1.46$) and brain slice ($n \sim 1.37$). The spatial resolution of the microscope was around 212 nm in x–y and 550 nm in z, as assessed by imaging fluorescent microspheres ('nominal spatial resolution'; SI Fig. 1).

To generate morphological contrast, we added a small but membrane-impermeant organic fluorescent dye (Calcein, 100 μM, Fig. 1b) to the ACSF bath solution in which the brain slices were submerged, as described before[5]. After optimizing the correction collar, we could acquire images of the tissue with high contrast and resolution (Fig. 1c), offering a macroscopic perspective of the anatomical layout replete with crucial structural details, such as dendrites and axons.

Despite the high density of the fluorescent label, it was possible to acquire confocal shadow images (COSHI) and z stacks at least 50 μm below slice surface (SI Movies 1, 2, 3), before image quality decreased due to out-of-focus blur, aberrations and light scattering.

Given the diffusional replenishment of dye molecules, it was possible to acquire a high number (>100) of time-lapse frames (Fig. 1d, SI Movie 4) with no signs of phototoxicity or drop in signal-to-noise ratio (SNR) (Fig. 1e, SI Fig. 2).

Next, we checked the utility of COSHI for studying microglia and their morphological dynamics in living brain tissue. Microglia are tissue-resident macrophages that are critical for the immune defense of the brain. They have highly branched and motile processes, which touch and may influence dendritic spines during brain development and neuroplasticity[14,15]. However, it remains challenging to determine which other anatomical structures they come into contact with in the surrounding neuropil.

To address this challenge, we used organotypic brain slices from transgenic mice (CX3CR1-EGFP) where microglia are highlighted by EGFP and labeled the ECS with a red dye (Alexa Fluor 594, 200 μM). Indeed, this combination made it possible to reveal the complex arborization of microglial processes and their anatomical context (Fig. 2A, SI Movie 5). We manually segmented cell bodies and larger processes in the COSHI images and identified physical contacts between EGFP-labeled microglial processes and unlabeled cellular structures (SI Movie 6). For example, one process of a microglia appears to wrap around a spiny dendrite from a CA1 neuron over several microns (Fig. 2B), while another process from the same cell neatly co-aligns with bundles of axons traversing the field of view (Fig. 2A, upper right panel). Systematic and quantitative insights into such cell-cell contacts based on unbiased observations will become increasingly possible with new computational image segmentation tools emerging in the field of connectomics, which would be very difficult to achieve with sparsely-labeled brain tissue.

Next, after inflicting a local laser lesion, which triggers a rapid and orchestrated immune reaction[16,17], it was possible to see how microglial processes navigate through the dense anatomical environment towards the lesion site (Fig. 2C, SI Fig. 3a, SI Movie 7). Microglial processes moved at different rates depending on local cellular structure. When cell bodies were present between the process tip and the laser lesion, microglial processes moved faster and trekked larger distances to home in on the lesion, compared to microglial processes that traveled through the neuropil (Fig. 2D, SI Fig. 3b; 0.07 μm/s for 17.1 μm through neuropil, 0.12 μm/s for 28.1 μm around cell bodies), possibly reflecting differences in physical hindrance or traction encountered by the microglial processes along the paths. Hence, the shadow imaging approach paired with positive labeling can provide insights into how cell motility and migration depend on local tissue architecture.

Microglia are highly phagocytic cells, particularly in pathological settings, where they are involved in clearing detritus in injured tissue. Phagocytosis is the engulfment of large substrates (>0.5 μm[18]) into cup-shaped cellular structures and thus is amenable to confocal imaging. The phagocytic cups of microglia are positioned at the end of their fine processes and are on average ~6 μm in diameter[19]. While they are known to engulf both dead or damaged as well as intact structures[20–22], how they triage interactions with these distinct targets in living brain tissue has remained difficult to assess.

Shadow imaging opens up a way to determine if microglial processes prefer phagocytosing lysed or relatively intact structures. Following the induction of the laser lesion, intact and lysed structures were identified by thresholding the inverted COSHI image (grayscale >70%, intact; grayscale <2%, lysed). An ROI indicating this developing lesion was outlined as a combination of lysed tissue (dye-positive) and swelling/blebbing intact tissue (dye-negative). By comparing the overlap of phagocytic cups with either lysed (dye-positive) or intact (dye-negative) structures in the developing lesion, we identified that

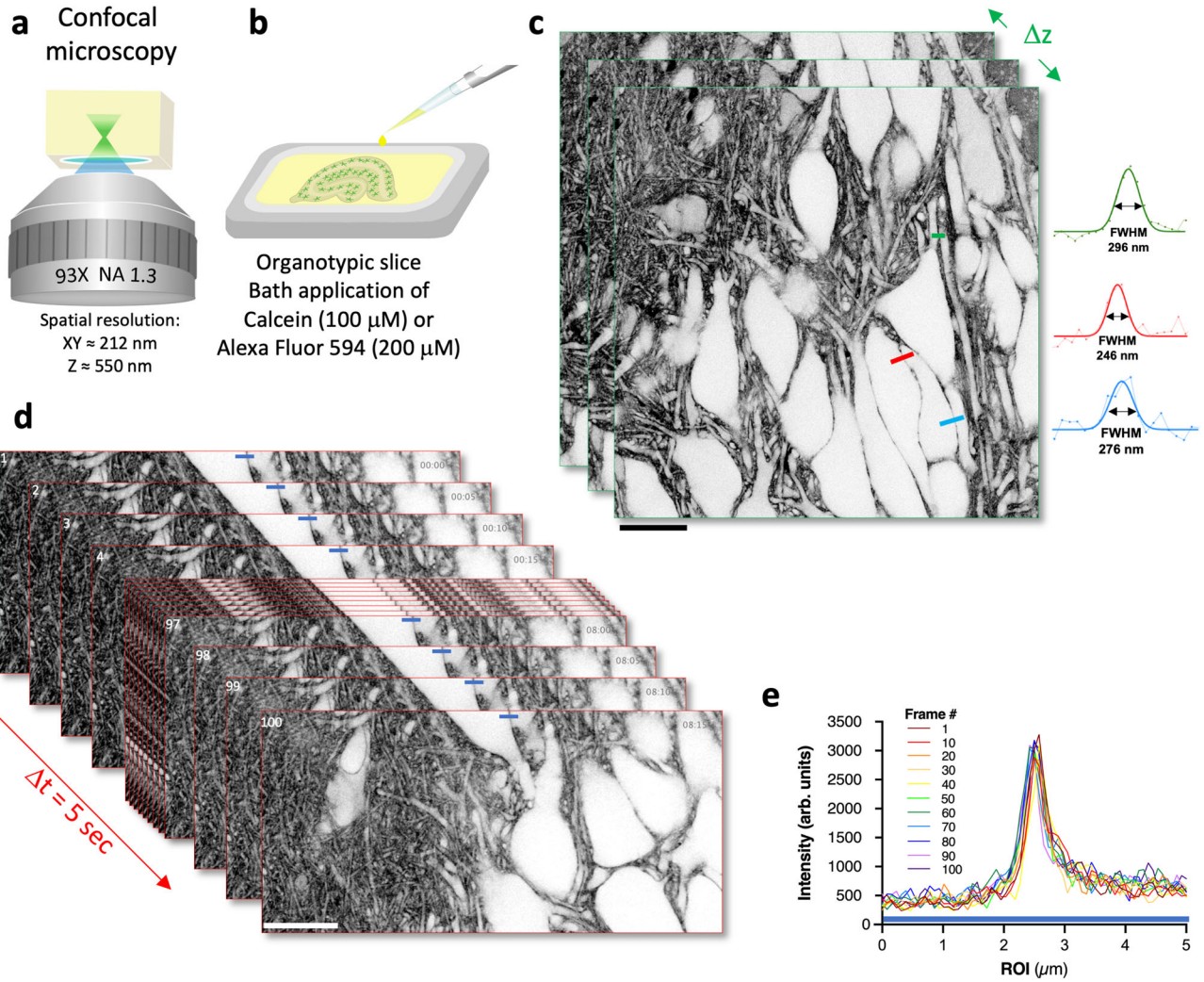

**Fig. 1 | Confocal shadow imaging in organotypic brain slices. a** Schematic of imaging technique based on a commercial inverted confocal microscope equipped with a motorized correction collar to reduce optical aberrations. **b** Organotypic brain slices were placed (upside down) into an imaging chamber containing the membrane-impermeant organic dye diluted in ACSF (100 µM of Calcein, or 200 µM of Alexa Fluor 594). **c** Z-stack of confocal shadow images (Calcein, 100 µM) of CA1 area of hippocampus in an organotypic brain slice (see also SI Movies 1, 2 and 3; representative of 5 independent experiments). Scale bar, 20 µm. Three examples of raw and Gaussian-fitted line profiles, color-coded with lines drawn on the image, indicating the spatial resolution in the tissue. FWHM: full-width half-maximum. **d** Time series of confocal shadow images (Calcein, 100 µM) of CA1 area of hippocampus in an organotypic brain slice, 1 frame every 5 s, 100 frames in total (see also SI Movie 4; representative series of 4 independent experiments). Scale bar, 20 µm. **e** Representative plot of line profiles across interstitial space between adjacent cell bodies (indicated by blue line) demonstrates no change in SNR across 100 consecutive frames (of 4 independent experiments).

---

microglia favor the phagocytosis of lysed structures over blebbing but non-lysed structures in response to a laser lesion (Fig. 2E, SI Fig. 4, 63.1% lysed, 30.8% intact, 6.1% both; see Methods for details).

This analysis reveals new aspects about the dynamic behavior of activated microglia in an 'emergency' situation, when they need to contain and clean up debris swiftly, while not producing too much collateral damage during such intense search and rescue operations.

As a point-scanning technique, it typically takes several seconds to acquire a single confocal image, which is often too slow for imaging $Ca^{2+}$ transients and other dynamic biochemical activities. In contrast, light-sheet microscopy reconciles high spatial with high temporal resolution because fluorescence excitation and detection are orthogonal to each other, enabling fast widefield imaging with little out-of-focus blur (Fig. 3a)[23].

To explore if shadow imaging is compatible with light-sheet microscopy, we used a custom-built microscope to image organotypic brain slices, where a thin and homogenous light sheet was created by a lattice excitation pattern and an oscillating mirror, as described previously[24,25]. The nominal spatial resolution of the lattice-light sheet (LLS) system was around 273 nm in x–y and 524 nm in z (SI Fig. 1).

Because of the efficient signal detection scheme in light-sheet microscopy, it was possible to use much lower dye concentrations (Calcein, 20 µM, Fig. 3a, b). We could acquire high-contrast light-sheet shadow images (LISHI) of the tissue, revealing fine details of its cellular microarchitecture across hippocampal layers (Fig. 3c), almost as well as COSHI but with >100X higher imaging speeds (Fig. 3d, SI Movie 8, SI Movie 9). To switch from the oblique light-sheet to a horizontal viewing orientation, the images were computationally de-skewed and rotated, effectively increasing the field of view (Fig. 3e).

The speed advantage makes it possible to perform LISHI alongside high-speed LLS imaging of functional signals such as glutamate release and astrocytic calcium transients. To demonstrate this possibility for popular green biosensors, we switched the dye for shadow imaging to Alexa Fluor 568 (SI Movie 10). Acquiring LLS images at a frame rate of 100 Hz, we could detect spontaneous glutamate signals in the neuropil using iGluSnFR expressed in neurons, and see their anatomical context in the horizontal LISHI images (Fig. 3f, g, SI

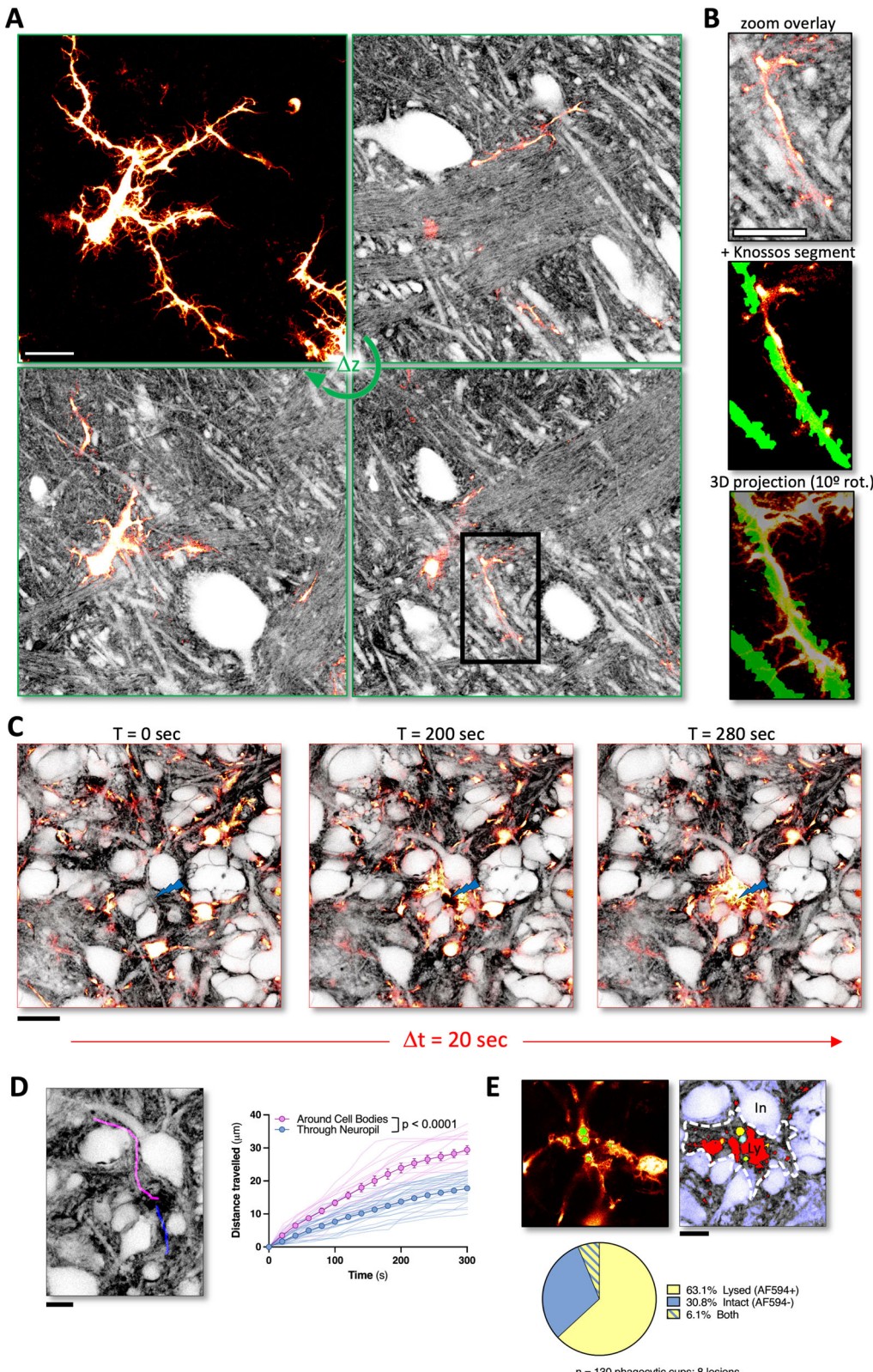

Movie 11, 12). Likewise, we could record short-lived astrocytic calcium signals using the biosensor GCaMP6 and place them into their anatomical context (Fig. 3h, i, SI Movies 13, 14).

2-photon microscopy is the main technique used for imaging acute brain slices or in vivo, offering superior SNR and optical sectioning deep inside light scattering tissue (Fig. 4a)[26]. However, shadow imaging in these popular experimental preparations poses unique challenges for fluorescence labeling and image contrast. Inevitably, there are many dead and cut open cells on the surface of an acute slice, which will take up the fluorescent dye if it is bath-applied, diminishing image contrast between cellular compartments and ECS.

To circumvent this problem, we spritz-injected the dye within brain slices (>50 µm below surface), where cells are mostly intact, via a pressurized patch pipette (Fig. 4b). Using a commercial 2-photon

**Fig. 2 | Microglial navigation through complex brain tissue is revealed by confocal shadow imaging. A** Z-stack of confocal shadow images (AlexaFluor 594, 200 μM) of CA1 area of hippocampus in an organotypic brain slice from a transgenic mouse line (CX3CR1-EGFP), where all microglial cells are fluorescently labeled with EGFP (see also SI Movie 5; representative of 4 independent experiments). Top left panel shows maximum-intensity projection of only EGFP-labeled microglia. Other panels show overlay of EGFP and shadow image for three different optical sections (16.5, 20.5, and 23 μm below the slice surface). Scale bars, 20 μm. **B** Zoom inset of overlay of EGFP and shadow image (20.5 μm depth), with web-Knossos-based manual segmentation of putative dendrite, and 3D projection (10° horizontal rotation) of segmented dendrite enwrapped by EGFP-positive microglial process (see also SI Movie 6). Scale bars, 20 μm. **C** Time series of confocal shadow images (AlexaFluor 594, 200 μM) showing labeled microglia cells (CX3CR1-EGFP) reacting to a laser lesion (blue bolt), rapidly extending their fine processes towards the site of the lesion (see also SI Movie 7; representative of 8 independent experiments). Scale bar, 20 μm. **D** Representative manual traces of EGFP-positive microglial processes extending towards the site of laser lesion, impeded by either cell bodies (magenta) or by neuropil (blue). Scale bar, 10 μm. Plot of mean distance traveled (μm; mean ± SEM) by these processes over time (seconds; $n = 29$ microglial processes through neuropil, $n = 14$ microglial processes around cell bodies, from 8 independent experiments, thin lines = traces of individual processes). Two-way repeated measures ANOVA with Bonferroni *post hoc* correction (time: $F_{(2.419, 99.19)} = 600.5$, ****$p < 0.0001$, process path: $F_{(1, 41)} = 58.01$, ****$p < 0.0001$, interaction: $F_{(15, 615)} = 37.23$, ****$p < 0.0001$). Source data are provided as a Source data file. **E** Representative image (left) from 20 min after lesion induction showing EGFP-positive microglial processes with phagocytic cups labeled (green). Scale bar, 10 μm. Representative COSHI image (right, grayscale) from the identical timepoint with overlay of intact (dye-negative; blue) and lysed (dye-positive; red) tissue, as well as phagocytic cups (green) and developing lesion area (white dashed line). Contents of phagocytic cups are identified by overlay (cyan−intact; yellow−lysed). Pie chart indicates the relative contents of phagocytic cups over 8 lesion time series ($n = 130$ phagocytic cups examined during 8 independent experiments). Source data are provided as a Source data file.

microscope with a 40X water-immersion objective (NA 1.0, spatial resolution x–y = 350 nm and z = 1.5 μm; SI Fig. 1), it was also possible to obtain shadow images with high contrast over large fields of view, temporarily lighting up the anatomical layout of the slices (Fig. 4c).

To increase and prolong fluorescence contrast, we used a dextran-conjugated dye (Alexa Fluor 488-Dextran, 500 μM), which slowed the dispersion of the dye in the ECS, making it possible to inject a relatively small volume (<1 μL) at low pressure with minimal tissue disturbance.

To demonstrate the ability of 2-photon shadow imaging (TUSHI) to reveal the anatomical context of a specific set of fluorescent cells, we spritzed the dye into acute brain slices prepared from mice implanted with YFP-labeled GBM tumor cells, which is a mouse model of glioblastoma of the mesenchymal subtype[27,28]. With two fluorescence detection channels (for YFP and Alexa Fluor 488), it was possible to image the proliferating tumor cells and observe their spatial integration in the tissue (Fig. 4d). Of note, this is the first study where the shadow imaging technique is applied to tumor tissue.

Taken together, the 'spritz-shadow imaging' technique, which is a variant of 'shadow-patching' for targeted electrophysiological recordings in vivo[29], can on the fly augment slice physiology studies with visual information on the anatomical environment.

Finally, we set out to extend the shadow imaging concept to the mouse brain in vivo to pave the way towards longitudinal neuroanatomical studies in mouse models of neuroplasticity and brain diseases.

We used a home-built 2-photon microscope with a 60X silicone oil objective (NA 1.3, WD 0.3 mm) that was equipped with a programmable spatial light modulator (SLM) to help reduce optical aberrations coming from the microscope itself and when imaging deeper inside brain tissue (Fig. 5a). The nominal spatial resolution of the microscope was around 320 nm in x–y and 925 nm in z (SI Fig. 1).

In addition to the usual challenges of in vivo imaging, such as brain motion and limited optical access, shadow imaging in vivo requires generating fluorescence contrast inside a large and sealed-off compartment. The labeling of the interstitial fluid should be strong, long-lasting, and minimally invasive. Instead of injecting the dye directly into the tissue, which proves to be too disruptive and unreliable, we performed stereotaxic injections into the ipsilateral ventricle controlled by a high-precision syringe pump, delivering the dye (4 μL of 50 mM of Alexa Fluor 488) over an extended period of time (~10 min) (Fig. 5b). This amount of dye theoretically corresponds to a concentration of a few millimolar in the interstitial fluid, if averaged over the entire mouse brain ECS. However, because of clearance, the effective concentration was likely much lower.

In this way, it was possible to achieve reproducible and strong brain-wide fluorescence ECS labeling that peaked shortly after the injection (<1 h) but persisted at workable levels for at least four hours (47.5 ± 2.2% drop from 1 to 4 h after injection; $n = 4$; Fig. 5c). The

procedure was optimized for the animals to recover and live on normally. While this labeling approach of the interstitial fluid is limited to one-off imaging sessions, new labeling strategies can be envisioned to permit repeated imaging over several days or weeks, for instance based on dye injections via a stably implanted cannula or genetically encoded markers that become fluorescent only in the ECS.

Following standard protocols for craniotomy and window implantation, leaving the dura mater intact, we could create stable and clear optical access to the mouse cortex. An anesthesia protocol based on intraperitoneal injections of ketamine and xylazine, and careful physical positioning of the animal under the microscope using ear-bars for head fixation prevented almost all motion artifacts from inspiratory muscle contractions, enabling image acquisitions with minimal motion blur.

Applying these technical measures, we could acquire high-contrast 2-photon shadow images of the superficial layers of somatosensory cortex (repeated in 15 animals; Fig. 5d). The resolution and contrast of the images were lower for in vivo than for organotypic brain tissue, which can be explained by the use of longer wavelength light 2-photon excitation (λ = 920 nm *versus* 488 nm), residual brain motion and presumably lower dye concentrations in vivo due to clearance from the interstitial fluid. Nevertheless, the TUSHI approach can reveal the outlines of neuronal cell bodies and blood vessels, where finer details like astrocytic endfeet, pericytes, and perivascular spaces (PVS) are readily discernible (Fig. 5d).

Of note, we could measure the width of the PVS around capillaries and blood vessels (Fig. 5e), which may play an important role in regulating brain fluid dynamics[30,31]. The average size of these spaces is around half a micron around capillaries (FWHM$_{PVS}$ = 0.58 ± 0.03 μm; FWHM$_{Capillary}$ = 3.7 ± 0.1 μm; N$_{capillary}$ = 105; N$_{PVS}$ = 210; N$_{mice}$ = 7; Fig. 5f), which is substantially larger than most interstitial gaps in the neuropil and thus may serve as an effective conduit for interstitial fluids. Shadow imaging in vivo thus makes it possible to quantitatively analyze this elusive component of the neuro-vascular system, and thereby to investigate the influence of its size and dynamics on the delivery and clearance of metabolites and biologicals in the brain. In addition, we observed a strong correlation (Pearson r = 0.838; $p < 0.0001$; N$_{capillary/PVS}$ = 79, N$_{mice}$ = 7; Fig. 5g) between the width of the PVS and the diameter of the associated blood vessel, suggestive of a functional relationship.

Moreover, by taking z stacks of up to 200 μm in depth, it was possible to survey sizable volumes of brain tissue (repeated in 15 animals; Fig. 6a, SI Movie 15). Because the dye is diffusible in the ECS, it was possible to acquire a high number of time-lapse images unaffected by bleaching (SI Fig. 2, *right panel*).

The volume fraction (VF) indicates the relative amount of ECS in the brain. It is an important structural measure of the compactness of

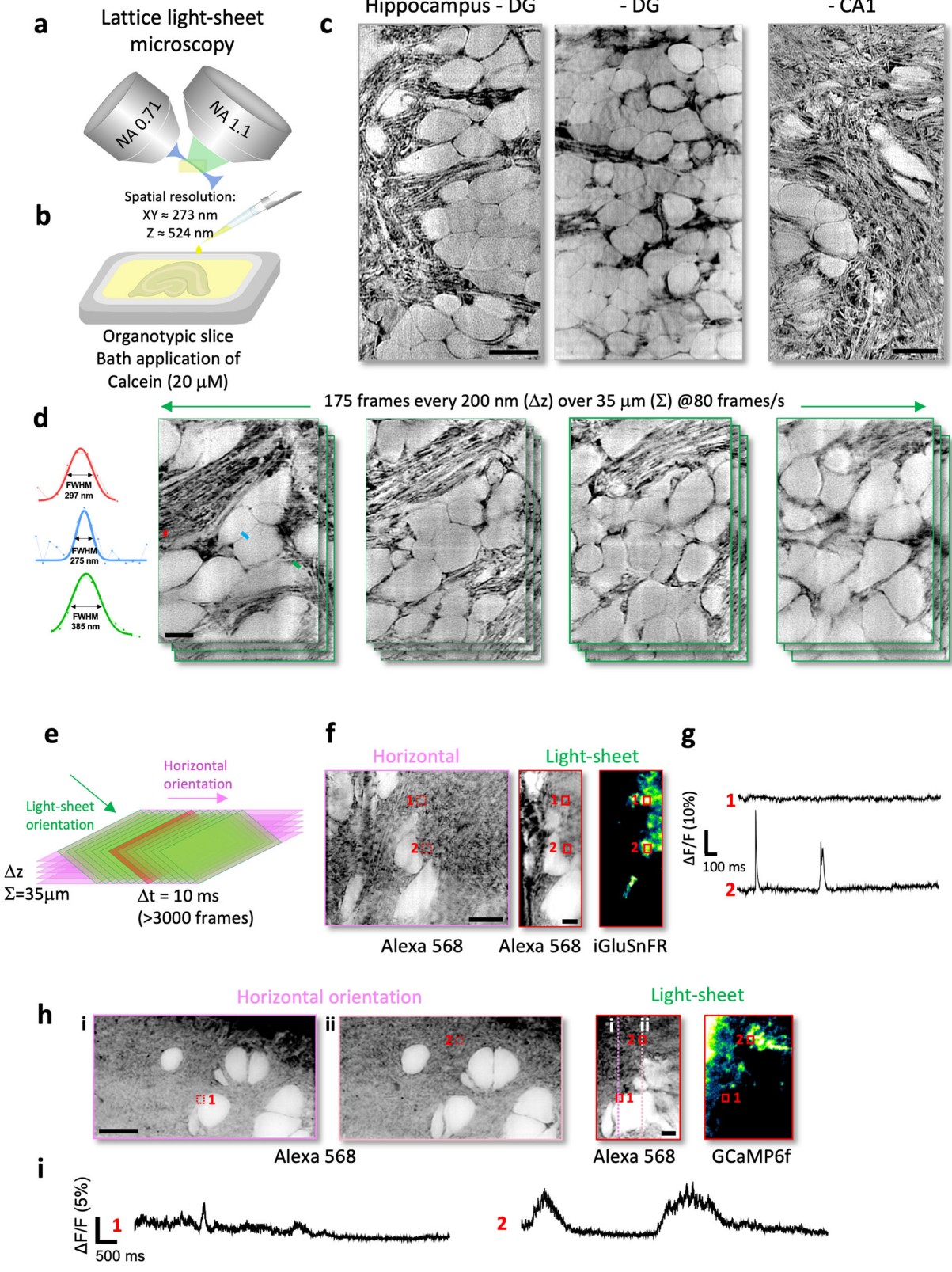

brain tissue, which exerts a strong influence on the spread and clearance of signaling molecules and metabolites[32,33]. While EM analysis based on cardiac perfusion of chemical fixatives has claimed for decades that the VF of the ECS is just a few percent, biophysical measurements suggest otherwise, estimating it to be about 20%. However, the biophysical approaches, which rely on monitoring over time the spread of chemical or fluorescent tracers released from a point source, can only yield VF estimates that represent averages over relatively large tissue volumes[34].

By contrast, shadow imaging opens up the possibility to estimate the VF directly and rapidly anywhere in a field of view at micron scale. As our spatial resolution was insufficient to separate ECS from cellular structures and segment the images faithfully, we used a method, which was originally developed to estimate the VF of sub-diffraction

**Fig. 3 | Light-sheet shadow imaging in combination with fast imaging of functional signals. a** Schematic of imaging technique based on a custom-built lattice light-sheet microscope. **b** Organotypic brain slices were placed (right side up) into an imaging chamber containing the membrane-impermeant organic dye diluted in ACSF (20 μM of Calcein). **c** Shadow images (representative of 6 independent experiments) acquired by light-sheet microscopy in different areas of the hippocampus in organotypic brain slices (see also SI Movies 8 and 9). Scale bars: 10 μm. **d** High-speed image z-stack (representative of 6 independent image acquisitions) in organotypic brain slice, 175 frames acquired at 80 Hz and a Δz step size of 200 nm. Scale bar, 5 μm. Left: Three examples of raw and Gaussian-fitted line profiles, color-coded with lines drawn on the image, indicating the spatial resolution in the tissue. FWHM: full-width half-maximum. **e** Schematic of two ways of visualization of 3D data acquired with the lattice light-sheet microscope: oblique light-sheet orientation or horizontal orientation, which is computationally generated by de-skewing, rotation, and image deconvolution. **f** Left: LISHI image viewed

in the horizontal orientation taken from a z-stack. Scale bar: 10 μm; Right: Images (representative of 11 independent image acquisitions) of the same region in Alexa-568 and iGluSnFR channels, both in the light-sheet orientation. Numbered red boxes indicate the ROIs of glutamate transients in the light-sheet orientation and their corresponding locations (dotted red boxes) in the horizontal view. Scale bar: 5 μm. **g** Time traces of averaged fluorescence (iGluSnFR) in ROIs. **h** Left: LISHI images viewed in the horizontal orientation taken from a z-stack. Scale bar: 10 μm; Right: Images (representative of 3 independent image acquisitions) of the same region in Alexa-568 and GCaMP6f channels, both in the light-sheet orientation. Numbered red boxes indicate the ROIs of calcium transients in the light-sheet orientation and their corresponding locations (dotted red boxes) in the horizontal view. The dotted pink lines (i, ii) indicate the sections in the corresponding horizontal LISHI z stack (i, ii). Scale bar: 5 μm. **i** Time traces of averaged fluorescence (GCaMP6f) in ROIs.

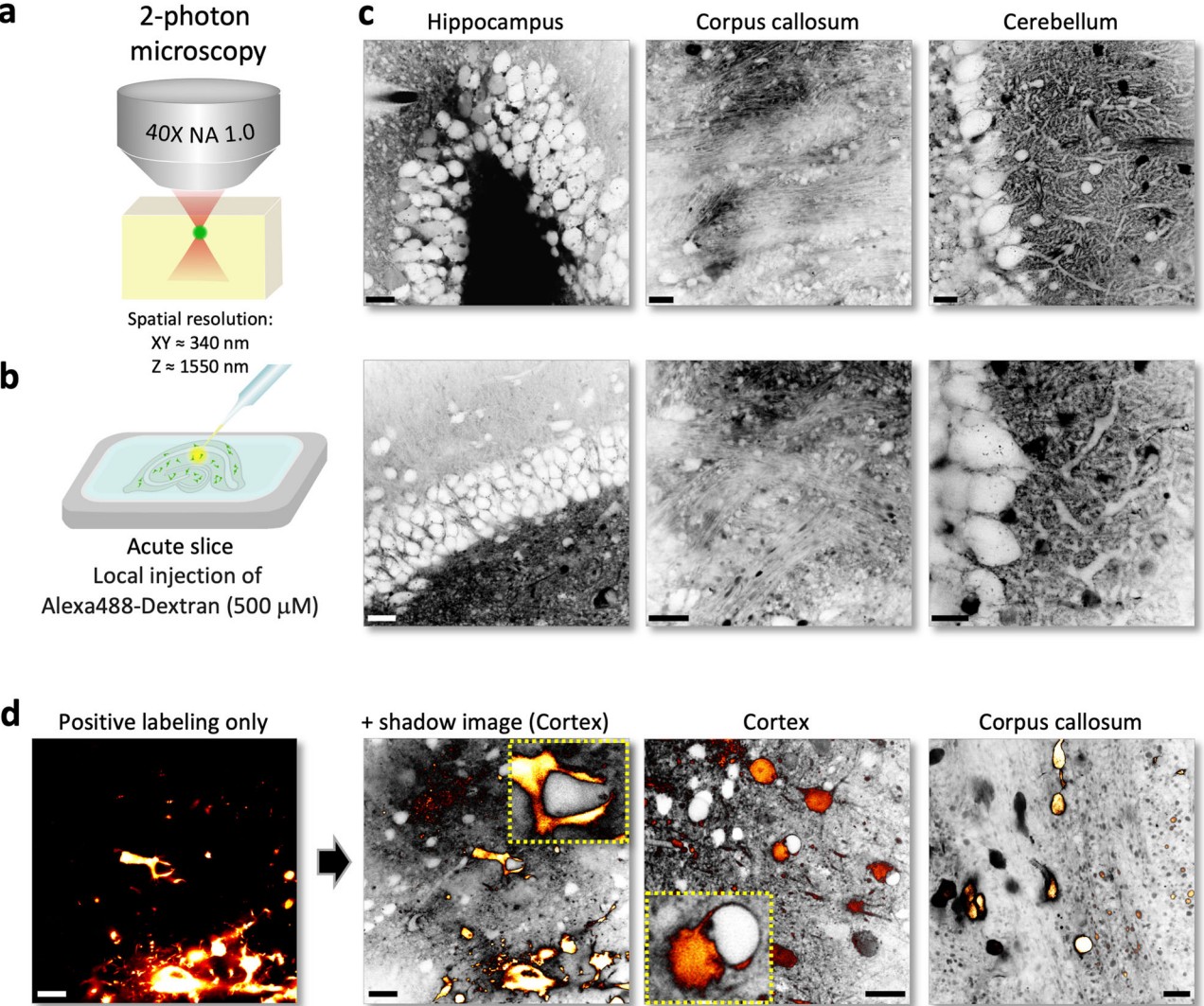

**Fig. 4 | 2-photon shadow imaging in acute brain slices. a** Schematic of imaging technique based on a commercial 2-photon microscope. **b** Acute brain slices were placed into the imaging chamber and a micro-manipulated patch pipette was used to inject the extracellular dye well below the surface of the slice (500 μM of Alexa488-Dextran). **c** 2-photon shadow images (representative of 15 independent experiments) of acute brain slices from different regions including hippocampus, corpus callosum and cerebellum, revealing their characteristic anatomical

structure. Scale bars, 20 μm. **d** Shadow imaging of YFP-labeled tumor cells in acute brain slices that were implanted into the brains of mice four weeks before slicing, revealing their anatomical relationship with the cellular environment. Examples (representative of 5 independent experiments) from cortex and corpus callosum. Scale bars, 20 μm. Note instances of intimate 'embrace' of unlabeled structures/cells by tumor cells.

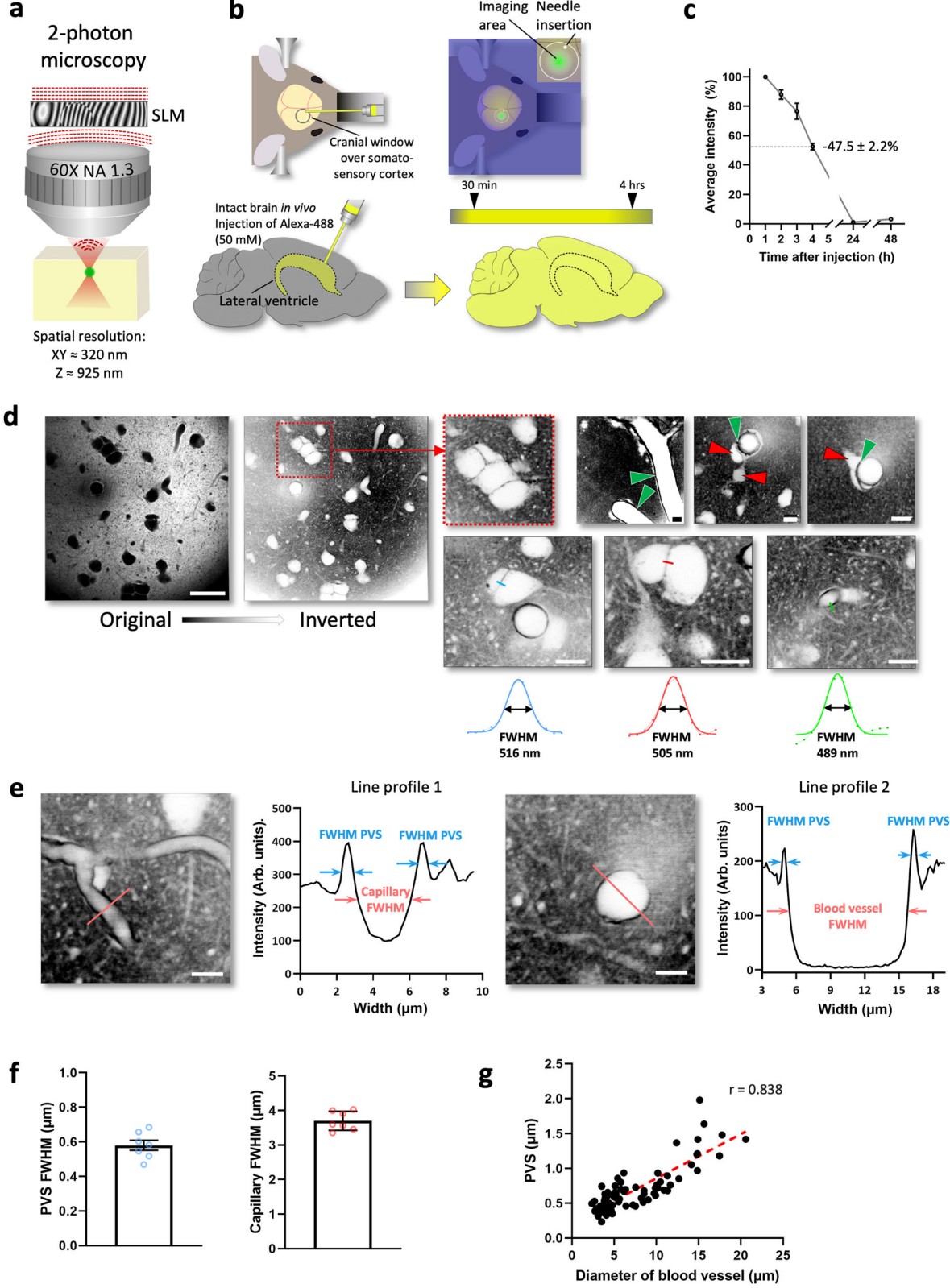

astrocytic processes[35] and is a priori independent of the spatial resolution of the microscope. It is based on calculating the average fluorescence intensity in the neuropil normalized by the brightest pixels in a given field of view, which approximate the signal from pure ECS (Fig. 6b). To get a good estimate of pure ECS, we chose regions of interest in the neuropil that contained blood vessels with perivascular spaces large enough for the PSF of our 2-photon microscope and

ensured that the photodetector operated well within its linear range. The VF values computed in this way are on average 23% ($23 \pm 0.7\%$ (mean ± SEM), $N = 4$ mice, $n = 36$ regions of interest; Fig. 6c).

Confirming the impression from the raw shadow images, our quantitative VF analysis indicates that the micro-anatomical architecture of brain tissue in vivo is organized in a relatively loose way with a high amount of ECS in the neuropil, consistent with biophysical VF

**Fig. 5 | 2-photon shadow imaging in the mouse brain in vivo. a** Schematic of imaging technique based on a custom-built 2-photon microscope equipped with a spatial light modulator (SLM) and a high-NA objective with a correction collar to reduce optical aberrations. **b** Schematic of the in vivo labeling strategy of the ECS, where a highly concentrated dye solution (<5 μL of 50 mM of Alexa Fluor 488) was stereotaxically injected slowly (over 10 min) into the ipsilateral lateral ventricle. **c** Time course of average fluorescence labeling intensity of the ECS in somato-sensory cortex after intraventricular dye injection. The fluorescence labeling appeared within 30 min and stayed elevated at workable levels for at least 4 h; the average intensity drop between 1 and 4 h was 47.5 ± 2.2% (mean ± SEM); $n = 4$. **d** Representative in vivo 2-photon shadow images acquired in the somatosensory cortex of an anesthetized mouse. Left: raw image, revealing cell bodies as large black 'shadows'; Scale bar, 25 μm; right: same image after inversion of the look-up-table; dotted square: zoomed-in part showing cluster of cell bodies; examples of shadow images delineating perivascular spaces (green arrows) running along blood vessels and putative pericytes and/or astrocytic endfeet (red arrows) wrapping around blood vessels. Scale bars, 10 μm. The imaging was repeated independently with similar results in 15 animals. **e** Perivascular spaces and blood vessel diameters were geometrically measured by line profiles laid orthogonally across the blood vessels (left: capillary; right: arteriole/venule), returning their widths (FWHM), scale bar, 10 μm. **f** Quantitative measurements of the diameters of perivascular spaces and blood vessels (mean ± SEM; $N_{capillary} = 105$; $N_{PVS} = 210$; $N_{mice} = 7$). Source data are provided as a Source data file. **g** The widths of perivascular spaces and the diameters of the blood vessels they enclose are highly correlated (Pearson correlation coefficient r = 0.838; $p < 0.0001$; $N = 79$). Source data are provided as a Source data file.

measurements[34] and completely at odds with the predominant EM view[36], where the neuropil appears extremely tight and compact.

To confirm that TUSHI in vivo can be combined with imaging positively labeled cells, we again turned to the mouse model of glioblastoma[27,28]. The approach showed YFP-labeled tumor cells infiltrating an area of neuronal cell bodies in motor cortex four weeks after their implantation into the mouse brain (repeated in 6 animals; Fig. 6d, SI Movie 16).

The technique may be valuable for studying various primary or secondary brain tumors in mouse models[37], how tumor cells interact with tissue microenvironment and impact brain structures[38]. It also is uniquely suited for ex vivo tissue samples where positive labeling is not an option, such as fresh human pathology specimen from the clinics, as a diagnostic tool or for monitoring therapy. In principle, TUSHI could be combined with any other form of positive cell labeling, i.e., circuit tracing, cell type reporters, or fluorescent biosensors to explore brain structure and function in view of the complete cellular context in living animals.

Taken together, our study presents a strong case for shadow imaging of living brain tissue using common diffraction-limited microscopy techniques. It can help reveal the anatomical context of labeled cellular players in a variety of patho-physiological settings, in addition to providing ECS structural information. In view of ongoing advances in ECS labeling, high-resolution microscopy, and computational tools, shadow imaging is poised to become an indispensable tool for live biological tissue studies.

## Methods
### Animals/brain tissue samples
**Regulatory issues.** Animal handling and experimental procedures were in accordance with the European Union and CNRS institutional guidelines for the care and use of laboratory animals (Council directive 2010/63/EU) and approved by the Institutional Animal Care and Use Committee at the University of Bordeaux, France (DAP 2019031909389750_v7).

**Mouse lines.** C57Bl/6J wild-type, CX3CR1-EGFP, and immunodeficient Rag 2 Gamma C-/- mice were used in this study. Mice were housed under a 12 h light/12 h dark cycle at 20–22 °C with *ad libitum* access to food and water in the animal facility of the Interdisciplinary Institute for Neuroscience and of Animalerie Mutualisée de Talence (University of Bordeaux/CNRS) and monitored daily by trained staff. All animals used were free of any disease or infection at the time of experiments. Pregnant females and females with litters were kept in cages with one male. We did not distinguish between males and females among the perinatal pups used for organotypic cultures, as potential anatomical and/or physiological differences between the two sexes were considered irrelevant in the context of this study.

**Organotypic brain slices.** Organotypic slice cultures were prepared according to the Muller method[12]. Briefly, hippocampal slices were obtained from postnatal day 5–7-old C57Bl/6J mouse pups. The animals were quickly decapitated, and the brains placed on cold sterile dissection medium (all in mM; 0.5 $CaCl_2$, 2.5 KCl, 2 $MgCl_2$, 0.66 $KH_2PO_4$, 0.85 $Na_2HPO_4$–$12H_2O$, 0.28 $MgSO_4$–$7H_2O$, 50 NaCl, 2.7 $NaHCO_3$, 25 glucose, 175 sucrose, 2 HEPES; all from Sigma). The hippocampi were dissected and sliced on a McIlwain tissue chopper to generate coronal brain slices of 350 μm thickness. After 20 min of incubation at 4 °C, the slices were transferred onto sterilized hydrophilic polytetrafluoroethylene (PTFE) membrane (FHLC04700; Merck Millipore) pieces, which were placed on top of cell culture inserts (Millipore, 0.4 mm; Muller method); the inserts were held in a 6-well plate filled with medium (50% Basal medium eagle (BME), 25% Hank's Balanced Salt Solution (HBSS), 25% Horse Serum, 11.2 mmol/L glucose and 20 mM glutamine; all from GIBCO) and cultured up to 14 days at 35 °C/5% $CO_2$. For CX3CR1-EGFP slices, culture medium was replaced with microglial-supportive growth medium (50% Basal medium eagle (BME), 36.5% Hank's Balanced Salt Solution (HBSS), 10% Horse Serum, 2% B27 plus supplement, 11.2 mmol/L glucose and 20 mM glutamine; all from GIBCO Culture) at DIV3-4. Culture medium was replaced every two days.

Organotypic slices were imaged in HEPES-buffered ACSF maintained at 34 °C or 37 °C (all in mM: 119 NaCl, 2.5 KCl, 1.3 $MgSO_4$, 1 $NaH_2PO_4$–$2H_2O$, 20 D-glucose, 1.5 $CaCl_2$, 10 HEPES; pH 7.4, 300 mOsm; all salts from Sigma).

**Acute brain slices.** Acute mouse brain slices were prepared (coronal and sagittal). Mice were anesthetized with isoflurane or ketamine/ xylazine (100/10 mg/kg) and perfused intracardially with ice-cold (for cortical and hippocampal slices) or warm (34 °C, for cerebellar slices) NMDG-based solution and then decapitated. NMDG-based solution contained (in mM): 92 NMDG, 2.5 KCl, 1.25 $NaH_2PO_4$, 30 $NaHCO_3$, 20 HEPES, 25 glucose, 2 Thiourea, 5 Na-ascorbate, 3 Na-pyruvate, 0.5 $CaCl_2$·$2H_2O$, and 10 $MgSO_4$·$7H_2O$. Extracted brains were placed in either ice-cold or warm NMDG-based solution to prepare 350-μm-thick slices using a vibrating microtome (Leica VT1200S, Leica Microsystems). Slices were transferred to NMDG-based solution at 34–36 °C for 15–30 min for initial recovery and then stored in a solution containing (in mM): 125 NaCl, 2.5 KCl, 10 glucose, 1.25 $NaH_2$ $PO_4$, 2 sodium pyruvate, 3 myo-inositol, 0.5 sodium ascorbate, 26 $NaHCO_3$, 2 $CaCl_2$, 1 $MgCl_2$ for 30 min at 34–37 °C and at room temperature afterwards.

For imaging, the slices were placed into a microscope chamber maintained at 32 °C where they were continuously perfused at a rate of 2 ml/min with an ACSF solution containing (in mM): 125 NaCl, 2.5 KCl, 11 glucose, 25 $NaHCO_3$, 1.25 $NaH_2PO_4$, 2 $CaCl_2$, 1 $MgCl_2$. The ACSF had an osmolarity of 295–300 mOsm, a pH of 7.4 and was continuously carbogenated (95% $O_2$ and 5% $CO_2$).

**Brains in vivo.** For in vivo imaging, we used 3–5-month-old C57B6/J or 3–4 months-old Rag 2 Gamma C-/- mice of both sexes. Animals were injected with buprenorphine (0.1 mg/kg) prior to the surgery for pain relief. Surgery was done under isoflurane anesthesia.

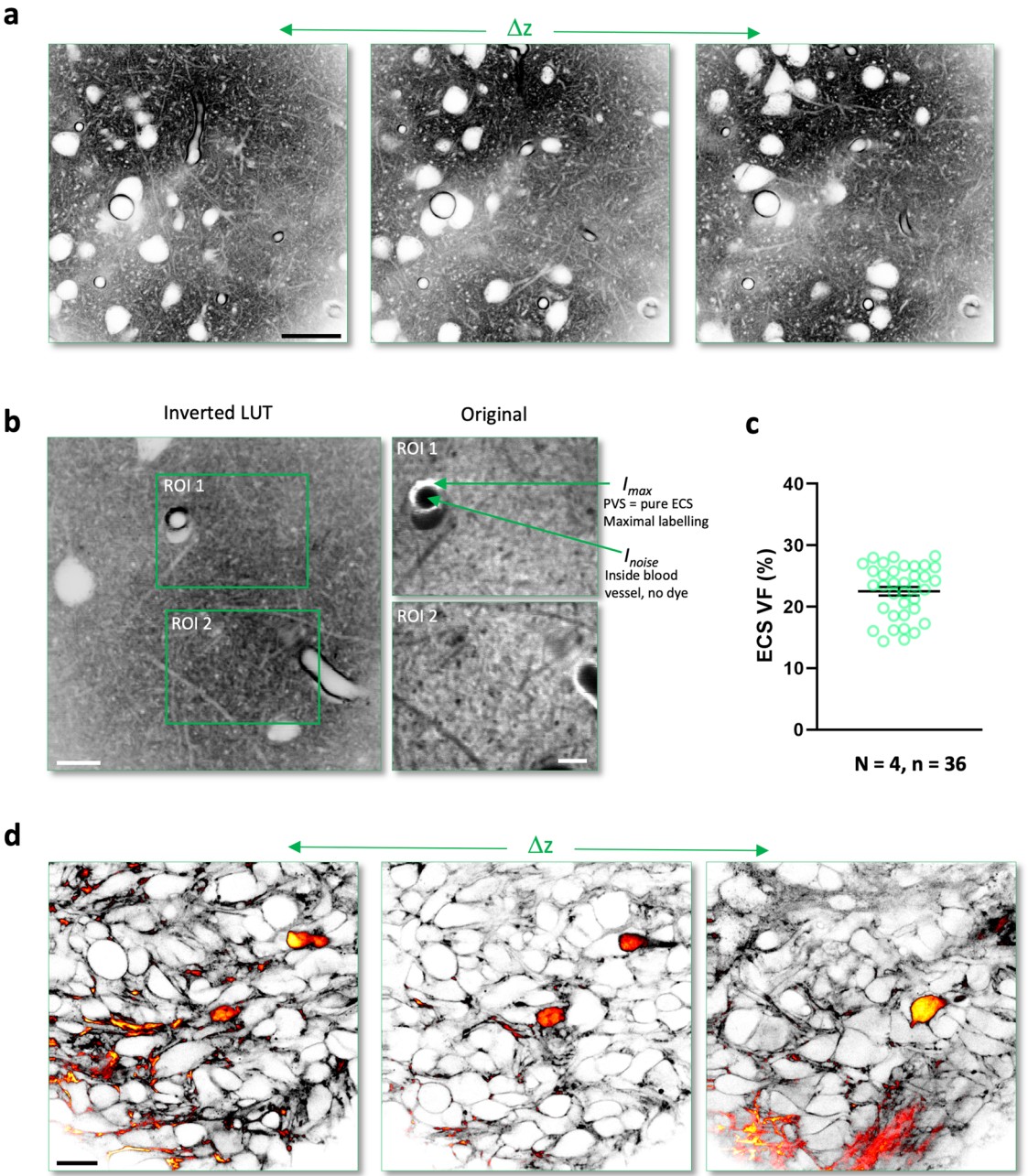

**Fig. 6 | 2-photon shadow imaging to estimate ECS volume fraction in vivo and reveal anatomical context of invading tumor cells. a** Z-stack of 2-photon shadow images acquired over >100 μm in depth (see also SI Movie 11). Scale bar, 25 μm; $N = 4$ mice; $n = 18$. **b** Regions of interest in neuropil analyzed for ECS volume fraction, based on normalized fluorescence intensity. Scale bars, 10 μm (left), 5 μm (zoom ins). **c** Group data of ECS volume fraction (VF) of cortical neuropil, based on normalized fluorescence intensity values, indicating that VF is substantial in mouse cortical neuropil in vivo, consistent with biophysical measurements in the literature. VF = $23 \pm 0.7\%$ (mean ± SEM); $N = 4$ mice; $n = 36$ regions of interest. Source data are provided as a Source data file. **d** In vivo 2-photon shadow images in motor cortex of mice implanted with YFP-labeled cells from a human GBM tumor cell line. The Alexa Fluor 488 (ECS) and YFP (tumor cells) fluorescence signals were spectrally detected. The images shown here were merged and pseudo-colored based on signal intensity (see also SI Movie 12). Scale bar, 20 μm. The imaging was repeated independently with similar results in 6 animals.

A round craniotomy (~4 mm in diameter), which left the dura mater intact, was made above somatosensory (C57B6/J mice) or motor M1 (Rag 2 Gamma C-/- mice) cortex. The dye (Alexa Fluor™ 488, carboxylic acid, Invitrogen, 4 μl with a concentration of 50 mM) was injected into the lateral ventricle on the side of the craniotomy at coordinates: M/L – 1.1, A/P – 0.52, D/V – 2.3 on the side of craniotomy at a rate of 1 μl/min using a motorized syringe pump. The craniotomy was covered with a glass coverslip (#1 thickness, diameter–4 mm) and sealed with glue and dental cement (Superbond C&B). After surgery, mice were anesthetized with ketamine/xylazine (100/

10 mg/kg) and placed on a heated blanket under the objective of a 2-photon microscope.

### Implantation of YFP-labeled glioblastoma cells
Orthotopic injections of 104 human YFP-positive GBM cells into 6–8-week-old immuno-deficient laboratory mice (Rag 2 Gamma C-/-) were performed using a stereotaxic frame at 2 mm on the left of the medial suture and 3 mm deep in the striatum.

The P3 primary GBM cell line was derived from a tumor patient with high-grade glioma[27]. After resection of a small piece of the tumor,

it was mechanically dissociated (Miltenyi) and grown in NBM media (ThermoFischer) supplemented with 2 mM l-glutamine, B27 supplement (ThermoFischer), 2 μg/ml heparin (Sigma-Aldrich), 20 ng/ml EGF (Peprotech) and 25 ng/ml bFGF (Peprotech), 100 U/ml penicillin (Sigma-Aldrich), and 100 μg/ml streptomycin (Sigma-Aldrich). The P3 cell line was transfected with a lentiviral plasmid FUGW-EYFP to express YFP in the cytosol (Addgene).

## Confocal microscopy

**Microscope.** Muller organotypic slices were imaged using a commercial confocal microscope (Leica-SP8-STED Falcon) with a 93X glycerol immersion objective (NA 1.3) equipped with a motorized correction collar (Leica Microsystems). The membrane support holding the slices was inverted and placed on a circular coverslip (18 mm diameter, # 1.5) in an imaging chamber (Ludin, Life Imaging Services), so that the slice directly faced the coverslip. A brass ring was placed on top of the membrane to prevent it from drifting. For wild-type slices, Calcein (Dojindo Laboratories) was diluted to a final concentration of 100 μM in the ACSF, while for CX3CR1-EGFP slices, AlexaFluor 594 carboxylic acid (Cat#A33082, ThermoFisher) was diluted to a final concentration of 200 μM. Images were acquired with a pixel size of 80 nm and pixel dwell time of 1.01 μs over a $125 \times 125$ μm$^2$ field of view (FOV). For slices labeled with Calcein, the pinhole was reduced to 0.5 Airy units (AU). For z-stacks, the step size was 0.5 μm, while for time series, the frame rate was 1 per 5 s for each $125 \times 62.5$ μm$^2$ FOV. For slices labeled with AlexaFluor 594, the pinhole was set at 1.0 AU. EGFP and AlexaFluor 594 emission was captured simultaneously using spectrally-separated detectors. A lesion was induced by focusing laser illumination to a 2.5 μm$^2$ region of interest for 1 s. Acquisition settings were reset to match baseline imaging parameters and subsequent images were acquired with a frame rate of 1 per 20 s for the $125 \times 125$ μm$^2$ FOV. The correction collar was set to a value that maximized image brightness and sharpness. Image acquisition was controlled by the microscope software (Leica Microsystems).

**Manual segmentation.** Large cellular structures in the COSHI z-stack were manually annotated and segmented using the WebKnossos platform[39] (SI Movie 6).

**Microglial lesion analysis.** EGFP-positive microglial process tips were labeled in consecutive frames (1 frame per 20 s) using the Manual Tracking FIJI plug-in (from baseline to time of arrival at the locus of laser lesion). These trajectory traces were then overlaid with the COSHI channel, to determine if there was an impeding cell body or neuropil only between baseline and lesion locations. Cumulative distance traveled to arrive at the lesion and the mean velocity (average between frames 3–7, linear portion of trajectory plot) were determined, comparing the presence of impeding cellular structures.

**Microglial phagocytic triage analysis.** COSHI time course images were spatially filtered using a median filter (4 pixels), then intact (dye-negative; pseudo-colored blue) and lysed (dye-positive; pseudo-colored red) cellular structures were identified by generating thresholded time-stacks based on the grayscale values of >70% and <2% for pixels in the inverted COSHI images, respectively (see SI Fig. 4). An ROI of the developing lesion area (white dotted line) was defined on a frame-by-frame basis as any area in the field of view with lysed or blebbing/dysmorphic intact structures.

To identify phagocytic cups, CX3CR1-EGFP time-stacks were binarized then duplicated. Holes were filled in one of two binarized microglia stacks using the Fill Holes FIJI function. The non-filled stack was then subtracted from the filled stack to generate a time-stack showing signal only where "holes" were present in the CX3CR1-EGFP images, identifying putative phagocytic cups. Using the 3D Object Analysis FIJI function weighted to favor more circular structures within

the size range of phagocytic cups, the regions of interest represented by these putative cups were redirected to the thresholded intact (dye-negative) or lysed (dye-positive) COSHI time-stacks to identify the phagocytosed contents. Enrichment of lysed or intact structures in phagocytic cups (SI Fig. 4) was demonstrated by comparing the lysed (red) and intact (blue) area in the developing lesion area (white dotted line) relative to their occurrence in phagocytic cups (green) over the duration of the time series. "In ROI" indicates the percentage of the developing lesion area containing lysed or intact structures as an average of all frames across each time series, whereas "In PC" indicates the percentage of phagocytic cups containing lysed or intact structures as an average of frames across each time series. A phagocytic cup was considered positive for intact or lysed structures if the median value of its contents was non-zero in either the dye-negative, dye-positive, or both channels, respectively.

## Light sheet microscopy

**Microscope.** Muller organotypic slices were imaged using a custom-built lattice light sheet microscope (LLSM), which was a replica of the setup described previously[24]. We used a dithered square lattice with an inner and outer NA of 0.44 and 0.55, respectively, giving a light sheet of constant thickness of approximately 0.6 μm over a length of 15 μm. Laser power incident on the illumination objective was controlled via an AOTF, and set to ~60 μW for 10 Hz and ~400 μW for 80 Hz acquisitions. The detection objective had an NA of 1.1 and a working distance of 2 mm. 3D volumes were acquired by a horizontal piezoelectric translator (step size 200 nm). A multiband emission filter (Semrock FF01-446/523/600/677) was used to block the light from the excitation lasers.

**LISHI.** During imaging, the slice was bathed in a heated imaging chamber containing a HEPES-based ACSF (at 37 °C) with 20 μM Calcein or Alexa Fluor 568 diluted in it. The camera frame rate was 10 Hz for Fig. 2C and SI Movie 8 and 80 Hz for Fig. 2D and SI Movie 9. Image acquisition was controlled by home-made software written in LabVIEW (shared by HHMI via research license agreement).

**LISHI and fast functional imaging.** Muller organotypic slices were introduced to either AAV5 pZac2.1 gfaABC1D-cyto-GCaMP6f (Addgene #52925) or AAV1 pAAV.hSynapsin.SF-iGluSnFR.S72A (Addgene #106176) via microinjections using a glass pipette connected to Picospritzer (Parker Hannifin). Briefly, the virus particles were puffed from a pipette positioned in the CA1 area of the slice by brief pressure pulses (30 ms; 15 psi). The injection was performed 1 day after slice preparation and the expressing slices were imaged two weeks after that. The 2D imaging of either glutamate or calcium signals was performed at 100 Hz speed using a 488 nm excitation laser. To remove the bleed-through from the LISHI channel into the iGluSnFR/GCaMP channel, a band-pass filter into the detection path was added (Semrock FF03-525/50). Following that, a z-stack of Alexa Fluor 568 signal was acquired using a 560 nm excitation laser (step size 200 nm).

## 2-photon microscopy (in acute brain slices)

Acute brain slices were imaged using a commercial 2-photon microscope (Prairie Technologies). For simultaneous two-color imaging of Alexa Fluor 488-Dextran (ECS) and YFP (tumor cells), the wavelength of the 2-photon laser (Ti:sapphire, Chameleon Ultra II, Coherent) was tuned to 920 nm, whereas for single-color imaging of Alexa Fluor 488, the wavelength was 850 nm. Images were acquired using a 40X water-immersion objective with an NA of 1.0 (Plan-Apochromat, Zeiss). Laser power was between 10 and 25 mW in the focal plane. The fluorescence signal was spectrally divided into two channels by a dichroic mirror with a cut-off wavelength of 514 nm and collected by PMT detectors. Images were acquired with a pixel size of 144 nm over a $295 \times 295$ μm$^2$ FOV and pixel dwell-times between 15 and 20 μs, corresponding to an

acquisition time of 62 s per image. Image acquisition was controlled by the microscope software (Prairie View).

For local dye injections, we used patch pipettes. The borosilicate glass capillaries were pulled with a vertical puller (Narishige, PC-10, Japan) to a tip resistance between 3 and 5 MΩ and filled with an ACSF solution containing 500 μM Dextran-Alexa Fluor 488 dye (10,000 MW, anionic, fixable, Invitrogen). The pipette tip was placed some 50 to 80 μm below the slice surface to inject the dye below the damaged and cut open cells on the surface of the acute slice. 8 to 10 psi of pressure was applied via a pressure generator (Picospritzer III, Parker) for several minutes at a time, which yielded a uniform distribution of the dye within the FOV, which persisted long enough to acquire images with high contrast between ECS and cellular structures.

## 2-photon microscopy (in vivo)

Imaging was performed using a custom-built 2-photon microscope, based on a standard commercial upright research microscope (BX51WI, Olympus). 2-photon excitation was achieved using a femtosecond mode-locked fiber laser (Alcor 920, Spark lasers) delivering <100 fs pulses at a wavelength of 920 nm and a repetition rate of 80 MHz. Laser power was adjusted using a Pockels cell (302 RM, Conoptics) to up to 30 mW power after the objective, depending on imaging depth and dwell time (normally 40 μs, but 200 μs for VF measurements).

The wavefront of the laser beam was modulated by an SLM (Aberrior Instruments) to correct optical aberrations induced by the optics of the microscope and the sample. Appropriate lens combinations were used to conjugate the SLM on a telecentric scanner (Yanus IV, TILL Photonics), which projected both scan axes on the back focal plane of the objective lens (UPLSAPO, 60X, silicone oil immersion, NA 1.3 Olympus) mounted on a z-focusing piezo-actuator (Pifoc 725.2CD, Physik Instrumente). The epi-fluorescence signal was descanned separated from incident beam using a long-pass dichroic mirror (580 DCXRUV, AHF) and detected by avalanche photodiodes (SPCM-AQRH-14-FC, Excelitas) with a bandpass filter (680SP-25, 520-50, Semrock) along the emission path. Signal detection and hardware control were performed with the Imspector scanning software (Abberior Instruments) via a data acquisition card (PCIe-6259, National Instruments).

## Assessing spatial resolution with fluorescent beads

Green fluorescent beads (diameter: 170 nm, Thermofisher) were dried on a coverslip and then mounted to a slide. $5 \times 5\,\mu m^2$ images were acquired on the respective microscopes under comparable imaging conditions. 5-μm-thick z-stacks were acquired with 0.2 μm steps and the section in focus was selected to determine xy resolution. In-focus beads were imaged in xz to determine axial resolution. Full-width at half-maximum (FWHM) measurements were done using the FWHM plug-in for FIJI.

## Statistics and reproducibility

LISHI images were first de-skewed, then deconvolved in 3D with an open-source Richardson Lucy (RL) algorithm (https://github.com/dmilkie/cudaDecon), using an experimentally measured PSF. The RL software is bundled into LLSpy, a python LLSM data processing and visualization toolbox (https://github.com/tlambert03/LLSpy). The deconvolution was run on a CUDA compatible GPU graphics card. A $3 \times 3$ median filter (noisy pixel removal) followed by background subtraction was applied before RL deconvolution with 20 iterations. Spurious horizontal and vertical stripes due to camera artifacts were removed with the ImageJ FFT filter. The deconvolved data were rotated to coverslip coordinates and resampled to obtain isotropic voxel dimensions $(102 \times 102 \times 102\,nm^3)$. Traces of glutamate (Fig. 3g) and calcium (Fig. 3i) transients were extracted from ROIs using ImageJ and ΔF/F was calculated with GraphPad Prism software.

TUSHI images were visualized using ImageJ. To estimate the volume fraction (VF) of ECS we used a normalized fluorescence intensity approach, which is a priori independent of the spatial resolution of the microscope. It is based on calculating the average fluorescence intensity in the neuropil normalized by the fluorescence signal from regions representing pure ECS such as perivascular spaces that were large enough to contain the 2-photon excitation spot, and thus provide a faithful measure of pure ECS. Data was analyzed in MS Excel and statistical analysis was done in Graph Pad Prism 9. All data are represented as mean ± SEM.

## Reporting summary

Further information on research design is available in the Nature Portfolio Reporting Summary linked to this article.

## Data availability

All data are available upon request. Source data are provided with this paper.

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

## Acknowledgements

The study was supported by grants from the European Research Council (ERC-SyG ENSEMBLE #951294), Human Frontiers Science Program (#RGP0036/2020), ERA-NET NEURON (ANR-17-NEU3-0005), Agence Nationale de la Recherche (ANR-17-CE37-0011) and Fédération pour la recherche sur le cerveau (FRC) to U.V.N. A.K.J.B. was funded by an Alberta Innovates Postdoctoral Fellowship, a EuroBio Imaging User Access Fund Award, and a Rebecca Hotchkiss International Scholar Exchange Award. A.I. was supported by a PhD fellowship from Bordeaux Neurocampus Graduate Program. S.K.K. was supported by a National Research Foundation of Korea grant (NRF-2019R1A2C2086052). M.A. was supported by JST FOREST Program (Grant Number JPMJFR2141, Japan). We thank our animal and cell culture facilities, as well as the Bordeaux Imaging Center (BIC) for technical support, R. Pullen for her help with the LLS imaging and L. Gerasimova for in vivo data acquisition and analysis.

## Author contributions

A.K.J.B., A.I., A.P.L., and Y.D. collected all imaging data. A.K.J.B., A.I., A.P.L., M.A., M.D., U.V.N., and Y.D. analyzed the data and prepared figures. A.K.J.B., A.I., A.O.A., A.P.L., G.L., J.G., L.M., K.O., M.A., M.D., M.S.F., S.B., S.K.K., T.P., U.V.N., and Y.D. provided technical input for sample preparation, labeling, and microscopy. K.O. and A.B. provided reagents. A.B. and R.J.T. provided postdoc supervision. U.V.N. conceived and supervised the study and wrote the paper with input from co-authors.

## Competing interests

The authors declare no competing interests.
