## [Peer Review File · Nature Communications]

Shadow imaging for panoptical visualization of brain tissue in vivoREVIEWER COMMENTS

Reviewer #1 (Remarks to the Author):

In a brilliant paper, Tonnesen et al (Cell 2018) previously demonstrated that the extracellular space of the brain tissue can be imaged with stunning clarity using shadow imaging and super-resolution STED microscopy. In the current manuscript, the authors aim to expand the utility of shadow imaging for the more conventional modalities (confocal, lightsheet, and two-photon microscopy) without super-resolution. Similar to the earlier paper, they show that the cellular architecture of the brain can be visualized in exquisite detail by labeling the extracellular space and digitally inverting the look-up table of the fluorescence images. While visually impressive, the current manuscript falls short however in a significant way. Lacking new technical advances, the authors need to present more substantial results with concrete examples of scientific insights that can be gained using the shadow imaging method. Instead, the authors repeatedly hinted at the possibilities without carrying through these lines of investigation. Examples are statements such as "possible to reveal the complex arborization of microglial processes amidst their fully visible anatomical context" (Lines 111-112) and "possible to see how microglial processes navigate through the dense anatomical landscape towards the lesion site" (Lines 114-115). Despite the motivation to image what cellular structures are contacted by the microglia (Lines 107-108), the authors made no attempt at analyzing such cell-cell contacts. Similarly, the authors stated that "light-sheet shadow imaging (LISHI) of cellular structures can ... be performed alongside high-speed imaging of biochemical dynamics such as synaptic release of glutamate and its spread in the ECS" without actually demonstrating the dynamic measurements that would have strengthened the manuscript.

A second major concern is the analysis of the volume fraction (VF). The binarized images using the two methods (Adaptive Thresholding and SpineJ) appear quite dissimilar and both are very different from the original images, raising concerns about what is actually being measured. Additional validation is needed to demonstrate the robustness of the VF analysis. The authors also need to discuss how the VF measurement is impacted by the optical resolution. Since the dimension of the extra cellular space (~40 nm) is below the diffraction limit, wouldn't these structures appear larger than their actual sizes (convolved with the

point-spread function), resulting in over-estimation of the VF? Please discuss.

Other concerns:

In vivo two-photon imaging of the brain tissue can routinely reach a depth of several hundred micrometers without adaptive optics, but here the authors reported z stacks of "up to 100 μm ". Please discuss what limits the imaging depth.

Reviewer #2 (Remarks to the Author):

Dembitskaya et al. described an extracellular fluorescence labeling method (shadow imaging) combined with three imaging modalities: confocal (COSHI), light-sheet (LISHI) and 2-photon (TUSHI). The same lab demonstrated super-resolution imaging (STED) with this extracellular space labeling previously (Tønnesen et al., Cell, 2018), termed SUSHI. Inheriting main advantages from SUSHI: unbiased labeling, reduced photobleaching, and potentially reduced toxicity, as compared with conventional fluorescence labeling approaches targeting at intracellular compartments, I find this technique interesting and will be useful for some biological applications. Demonstrating the combination of extracellular fluorescence labeling with multiple commonly used imaging modalities (COSHI, LISHI and TUSHI) will be valuable to the communities of both microscopy and neuroscience.

My major concerns about the current manuscript are:

- A. What are the technical advances compared with the previously reported method?
- B. In this current work, super resolution was compromised for an enlarged imaging FOV. However, the knowledge gained from extended imaging volumes was not clear.
- C. Some statements and conclusions made in the manuscript were not well supported by imaging experiments.
- D. Some important experiment details were missing.

Please find my detailed questions and comments below:

1. For all imaging modalities applied in this work, their spatial resolutions were reported but all were indicated "data not shown". In online Methods, the authors described how they

assessed spatial resolution using sub-diffraction fluorescence beads. I would suggest the authors include their measurement results (i.e., measured point spread functions) as supplementary information.

2. The resolution measured from fluorescent beads does not represent the effective resolution supported by the “shadow imaging” method. Multiple factors would affect the practical resolution: extracellular labeling associated background and noise, sample induced aberrations, and inverted contrast processing. The effective resolution is an important parameter of “shadowing imaging” and should be carefully characterized. I would suggest the authors perform quantitative analysis, for example, show line profiles, or show the spectral representation of the images.

3. Line 101-102: the authors claimed that “it was possible...with little or no drop in signal-to-noise ratio (SNR) or signs of phototoxicity” – with “data not shown”. I would suggest the authors quantify image SNR and analyze how image SNR change during image acquisition. Also, how did the authors evaluate phototoxicity? Can the authors provide quantitative analysis of photo bleaching (e.g., signal change during image acquisition)?

4. Fig 1F: I would suggest the authors show images from both channels as well as the merged, so that microglial morphology can be better visualized together with neurons. I am curious here that the video shows cell body shrinkage (the neuron on the left of the laser lesion) upon laser lesion – may the increased local fluorescence be autofluorescence from tissue debris? From morphology, I think the bright yellow signal came from microglia, but I am curious that why microglia in the lesion area displayed dark hollows inside the cell bodies? These hollows were not observed in Fig. 1E.

5. Line 42-43, the authors claimed that electron microscopy and magnetic resonance imaging “cannot be combined in situ with other powerful neuro-technologies, such as Ca²⁺ imaging, electrophysiology and optogenetics.” This is not true, for example: (1) Lake et al., “Simultaneous cortex-wide fluorescence Ca²⁺ and whole-brain fMRI”, Nature Methods, 2020, (2) Chen et al., “MRI-guided robotic arm drives optogenetic fMRI with concurrent Ca²⁺ recording”, Nature Communications, 2019, and others.

6. Line 119-120: point-scanning techniques... “often too slow for imaging Ca²⁺ transients and other dynamic biochemical activities”, which was not demonstrated with LISHI, either. The authors only stated that “in principle LISHI can be performed alongside high-speed imaging of ...” Real calcium imaging should be performed to make this conclusion. If the

authors only demonstrated imaging with higher speed but not real functional imaging, I would suggest the authors tone down their conclusion here.

7. For in vivo TUSHI, 8 μ L of dye was injected, which is a very large volume of injection. As a comparison, GCaMP virus transfection only requires 30~50 nL per injection site. I

understand that the authors injected the dye not directly to the brain tissue but ipsilateral ventricle. But I wonder how invasive this procedure is and how practical this method is for longitudinal studies.

8. Supposedly the dye only labels extracellular space and does not go into the cells. But in Fig. 2G, there are some blobs inside cell bodies showing contrast – why is that? These blobs were not observed in Figs 1, 3.

9. The home-built 2-photon microscope was equipped with an SLM to reduce aberrations when imaging deep inside brain tissues. The objective working distance is only 0.3 mm. How deep did the authors image? Was aberration correction necessary in this case?

10. Line 107-108: the authors mentioned that “However, it remains unclear, which other cellular structures they come into contact with in the surrounding neuropil.” The readers would assume following experiments would answer this question; however, I did not see a conclusion drawn from COSHI imaging of microglia.

11. Line 26-27: the authors stated that “current neuroanatomical imaging approaches either require tissue fixation, do not have cellular resolution or only give a fragmented view”. Did the authors want to say: “..., either do not have cellular resolution or only give a fragmented view”?

12. What are the depths of the two images shown in Fig. 1C?

13. I believe Ref 15: (SPIE, San Francisco, France, 2019) should be California, USA.

Reviewer #3 (Remarks to the Author):

The authors extend a method of staining the extracellular space they introduced in 2018 for STED microscopy to diffraction-limited light microscopy in brain slices (confocal- and light-sheet microscopy) and in vivo (two-photon). As in their previous papers (Cell 2018, Glia 2021), they show that after contrast inversion, the shadow image can be combined with cell-specific fluorescent labels (of different color) to provide information about the position of the labeled cells in the tissue and neighboring structures. Using microglia and injected

fluorescent tumor cells as examples, movements of these cells can be tracked by time-lapse shadow imaging. Their customized two-photon microscope is equipped with SLM wavefront correction, resulting in excellent visualization of minute details such as perivascular spaces and astrocytic endfeet. An interesting methodological detail is the injection of dye into the lateral ventricle, which provided sufficient staining of the extracellular space for several hours after cranial window implantation. The technique is therefore not suitable for repeated imaging over days or during behavior.

The manuscript provides convincing images and movies to highlight the possibilities of shadow imaging in vivo, but does not deliver any new biological insights. Interactions between microglia and implanted tumor cells would be very interesting to study, but the current manuscript stops short of combining the two application examples. On the positive side, methodological improvements are described in sufficient detail to make the paper a useful resource for labs that would like to implement the technique.

Response to reviewers' comments

We thank the reviewers for their constructive feedback. Altogether, the reviews prompted and helped us to upgrade the study and to have a real impact on the fields of bio-imaging and neuroscience.

To this end, we have added new experiments, data and analyses, expanding the number of figures and the text accordingly, while striving to keep the manuscript as concise and accessible as possible.

Enabled by the shadow imaging approach, we were able to add multiple exciting biological observations to the revised version of the manuscript, concerning 1) the dynamic interaction of microglial processes with their environment, which should interest the large community of glial biologists, and 2) the neuro-vascular unit, which will be of high interest for experimentalists and modelers working on the blood-brain barrier, brain fluid dynamics, and vasoconstriction and vasodilation processes. Moreover, we show that shadow imaging can be carried out in conjunction with fast light-sheet imaging of biochemical activities, which should be of high interest for researchers studying the structure and function of neural circuits.

Altogether, we now demonstrate clearly that this new shadow imaging paradigm can readily open up a world of possibilities for neuroscientists to make discoveries about (sub-)cellular structures and their dynamic relationships with the surrounding environment under various (patho-)physiological settings.

In the following, we address the Reviewers' comments (marked in blue) in a point-by-point fashion, explaining and indicating all the changes/additions in the revised manuscript.

Reviewer #1 (Remarks to the Author):

In a brilliant paper, Tonnesen et al (Cell 2018) previously demonstrated that the extracellular space of the brain tissue can be imaged with stunning clarity using shadow imaging and super-resolution STED microscopy. In the current manuscript, the authors aim to expand the utility of shadow imaging for the more conventional modalities (confocal, lightsheet, and two-photon microscopy) without super-resolution. Similar to the earlier paper, they show that the cellular architecture of the brain can be visualized in exquisite detail by labeling the extracellular space and digitally inverting the look-up table of the fluorescence images. While visually impressive, the current manuscript falls short however in a significant way. Lacking new technical advances, the authors need to present more substantial results with concrete examples of scientific insights that can be gained using the shadow imaging method.

Instead, the authors repeatedly hinted at the possibilities without carrying through these lines of investigation. Examples are statements such as "possible to reveal the complex arborization of microglial processes amidst their fully visible anatomical context" (Lines 111-112) and "possible to see how microglial processes navigate through the dense anatomical landscape towards the lesion site" (Lines 114-115).

We thank the reviewer for the positive assessment of the shadow imaging approach, and prompting us to extract more biological insights from it, which we have managed to do in several ways.

We have now added new data and analyses to report on microglial motility towards the site of a laser lesion. We can see clear differences in the speed and distance traveled by microglial processes depending on the route they take through the dense tissue. Time-lapse shadow imaging shows that the tips of microglial processes travel faster along cell bodies than through the neuropil. We have added these data in **Fig. 2D** and **SI Fig. 2A** and **B**. This is an original observation, uniquely enabled by the shadow imaging approach. It should be of high interest for researchers studying cell motility and migration, e.g., in the context of brain development, inflammation and tumors, where tissue microarchitecture is likely to play an influential role.

In addition, we have examined which cellular structures the microglial processes touch and end up phagocytosing after laser lesions. The time-lapse shadow images reveal that microglia favor lysed structures (i.e., that are dye-positive), engulfing them in phagocytic cups, but sometimes also ingest

degenerating/blebbing structures (dye-negative), albeit less frequently. We have added these data in **Fig. 2C** and **SI Fig. 2D**.

This analysis brings to light interesting aspects about the complex and dynamic behavior of phagocytic microglia in an 'emergency' situation, when they need to remove toxic debris without inflicting too much collateral damage.

Notably, the shadow imaging approach was crucial here to give an unbiased view of all cellular structures. The physical distinctions that needed to be made involve readily recognizable domains (cell bodies versus neuropil) or swollen cellular structures, which confocal shadow imaging (COSHI) has no problems detecting and resolving.

In addition, we now present quantitative measurements on the size of perivascular spaces surrounding capillaries, which has never been done in the living brain (**Fig. 5E**). These sub-micron compartments could play an important role in brain fluid dynamics and be a part of a 'glymphatic system', where brain fluids are thought to pass from the perivascular space around arterioles into the ECS of the neuropil and are drained back into the perivascular space around venules. Shadow imaging in vivo thus makes it possible to study this elusive and potentially important part of the neuro-vascular system. In addition, we report a strong correlation between the width of the perivascular spaces and the diameter of the associated blood vessel (**Fig. 5G**), suggestive of a functional relationship. This observation can be also useful for the study of vasoconstriction/dilation.

Despite the motivation to image what cellular structures are contacted by the microglia (Lines 107-108), the authors made no attempt at analyzing such cell-cell contacts.

We have now tried to assess the contacts between the labeled microglial processes and the surrounding inversely labeled cellular structures such as neuropil vs. cell bodies, and dendrites vs. axons. Since this analysis is very labor-intensive, requiring painstaking reconstructions of the dense tissue, we demonstrated in an exemplary fashion that such anatomical distinctions for cell-cell contacts can be made (**Fig. 2B** and **SI Video 4**). A systematic and complete analysis is out of the scope of this study but it should become more feasible with new computational image segmentation tools that are emerging in the field of connectomics.

Similarly, the authors stated that "light-sheet shadow imaging (LISHI) of cellular structures can ... be performed alongside high-speed imaging of biochemical dynamics such as synaptic release of glutamate and its spread in the ECS" without actually demonstrating the dynamic measurements that would have strengthened the manuscript.

We have taken this criticism onboard and now demonstrate high-speed light-sheet imaging using biosensors for local glutamate (iGluSnFR) and astrocytic calcium (GCaMP6) signals and correlate them with the shadow imaging. We show that it is feasible to associate the anatomical and rapid functional imaging with light-sheet microscopy, which paves the way to study activity patterns at the circuit level with the benefit of seeing anatomical context. The new data sets are now included in the manuscript (**Fig. 3F, G, H and I**) and as videos in the supplemental information (**SI Video 9A, B and 10A, B**).

A second major concern is the analysis of the volume fraction (VF). The binarized images using the two methods (Adaptive Thresholding and SpineJ) appear quite dissimilar and both are very different from the original images, raising concerns about what is actually being measured. Additional validation is needed to demonstrate the robustness of the VF analysis.

We have considered this concern and have applied another approach to estimate the volume fraction (VF), based on a method originally introduced to estimate dynamic changes in the VF of fluorescently labeled astrocytes (Henneberger et al, Neuron 2020). In our case (VF of the extracellular space), the method works by comparing the fluorescence intensity levels between the 'grey zones' in the neuropil, where the (extra)cellular structures are not fully resolved, and the 'black

and white zones' that are either well inside cellular structures (like cell bodies) or outside cellular structures (like peri-vascular spaces, which are visible in the shadow images).

By first approximation, the measured fluorescence intensity scales with the volume of the ECS that is accessible to the dye within the detection volume (i.e., the PSF of the microscope). Hence, by dividing the signals in the grey zones by the signals from regions that represent pure ECS in the field of view, it is possible to estimate the VF of ECS regions with structures beyond the spatial resolution of the microscope. This approach is a priori independent of the microscope's spatial resolution, unlike the binarization-based approach.

Performing this calculation, we obtained average VFs of around 33%, which is slightly higher than the VF estimates based on image binarization, but confirms the overall range. This approach will overestimate the VF, if the regions we considered as pure ECS (normally, the brightest pixels within the peri-vascular spaces) contain some cellular structures and thus do not represent 100% ECS. Also, sub-linear detection of the fluorescence photons in the very bright regions of pure ECS would lead to an overestimate. However, we have tried to minimize both potential pitfalls by analyzing regions of interests that included larger pools of ECS (around blood vessels) and deeper sections where detector saturation should not be an issue.

Moreover, a simple geometric model of neuropil architecture, stacking up cylindrical axons (of 200 nm diameters) with a minimal distance of 20 nm, returns a theoretical ECS VF of 30%. In cell body layers the VF tends to be lower, presumably because the space can be occupied more completely by polymorphic cell bodies than by axons with simple cylindrical geometries. Consistently, the VF is around 32% in the neuropil (stratum radiatum) as compared to around 21% in the cell body layer (stratum pyramidale) in organotypic hippocampal slices measured with STED-SUSHI and based on normalized fluorescence (data not shown).

Altogether, our shadow imaging approach provides strong confirmation of the biophysical work by Charles Nicholson and associates, who have long argued that the VF of brain ECS is much higher than what EM analysis based on cardiac perfusion and chemical fixation claimed for decades. In EM images the ECS appears nearly depleted, whereas the *in vivo* shadow images show (qualitatively and quantitatively) that the *in vivo* reality is very different. Notably, our approach does so in a direct imaging way at micron-scale, unlike the biophysical measurements, which rely on modeling the spread of tracers released from a point source and averaged over large chunks of brain tissue (hundreds of microns across).

Having said that, the analysis we present here is far from perfect. Fortunately, both approaches, based on segmentation and normalized fluorescence intensities can get more accurate and reliable, as the spatial resolution of the microscopic approach becomes higher. The former will benefit from better resolved ECS structures and the latter from more accurate measurements of what constitutes pure ECS fluorescence signals.

In the revised manuscript (**Fig. 6B2, C**), we present the new data and explain the caveats of the different approaches.

The authors also need to discuss how the VF measurement is impacted by the optical resolution. Since the dimension of the extra cellular space (~40 nm) is below the diffraction limit, wouldn't these structures appear larger than their actual sizes (convolved with the point-spread function), resulting in over-estimation of the VF? Please discuss.

Indeed, the diffraction blur is expected to over-estimate the VF based on segmentation. The fact that our *in vivo* VF estimates are consistent with the biophysical studies and with our own previous measurements using STED-SUSHI in organotypic hippocampal slices may be explained by the fact that pixels get assigned either to the ECS or non-ECS fraction at a threshold midway between the darkest and brightest pixels, effectively limiting the effect of image blur due to diffraction. While the results are consistent, it remains unsatisfying to rely on segmentation/binarization of diffraction-

limited images of sub-diffraction structures. For this reason, we also estimated the VF using the method of normalized fluorescence intensities, which is a priori independent of the optical resolution. The VF estimates based on this normalized approach are even slightly higher than the estimates based on binarization, providing additional evidence that the ECS is relatively voluminous in the neuropil *in vivo*, in agreement with the biophysical measurements.

Altogether, the shadow approach presents a promising step towards reading out the ECS VF at micron-scale in the living brain, which is expected to improve with better resolution and image analysis algorithms.

Other concerns:

In vivo two-photon imaging of the brain tissue can routinely reach a depth of several hundred micrometers without adaptive optics, but here the authors reported z stacks of "up to 100 μm ". Please discuss what limits the imaging depth.

The relatively shallow depth comes from the short working distance of the silicone oil objective (300 μm) and our confocal detection scheme, which makes it hard to image deeper because of scattering of the fluorescence photons and signal loss. The reasons for not using non-descanned detectors in our setup are historical because the setup was designed for STED imaging close to tissue surface (<100 μm), using avalanche photodiodes as detectors, which are superior to PMTs when scattering is not dominant. For both of these issues (working distance and confocal detection) there are technical solutions, which leave room for future improvement.

Reviewer #2 (Remarks to the Author):

Dembitskaya et al. described an extracellular fluorescence labeling method (shadow imaging) combined with three imaging modalities: confocal (COSHI), light-sheet (LISHI) and 2-photon (TUSHI). The same lab demonstrated super-resolution imaging (STED) with this extracellular space labeling previously (Tønnesen et al., Cell, 2018), termed SUSHI. Inheriting main advantages from SUSHI: unbiased labeling, reduced photobleaching, and potentially reduced toxicity, as compared with conventional fluorescence labeling approaches targeting at intracellular compartments, I find this technique interesting and will be useful for some biological applications. Demonstrating the combination of extracellular fluorescence labeling with multiple commonly used imaging modalities (COSHI, LISHI and TUSHI) will be valuable to the communities of both microscopy and neuroscience.

We thank the reviewer for the positive assessment of our study.

My major concerns about the current manuscript are:

A. What are the technical advances compared with the previously reported method?

The paper offers a number of solutions / innovations regarding the labeling, which is central to the shadow imaging concept. In addition, a key technical insight is basically the demonstration that even non-super-resolution microscopy modalities can be used and are capable of extracting interesting biological information. We propose and validate several ways to do shadow imaging, which are readily adoptable by a large user community, and not restricted to the relatively small number of labs with access to super-resolution STED microscopy.

Specifically, we came up with effective methodologies to label the ECS in organotypic and acute slices as well as in the brain *in vivo*, with high contrast and little toxicity, and established, which dye combinations and concentrations are useful for positive/negative color contrast.

This was not a trivial task, and took much effort and time to be checked and optimized. Altogether, there is not a singular technical advance, but rather a series of incremental improvements, which were necessary to reach the final quality level and repeatability that make our study worthy of attention and further pursuing.

B. In this current work, super resolution was compromised for an enlarged imaging FOV. However, the knowledge gained from extended imaging volumes was not clear.

Achieving microscopic resolution over large FOVs is crucial for anatomical imaging, given the multi-scale sizes of the anatomical structures and their physical relationships. The multi-scale view is a central strength of light microscopy. Indeed, in this study, we relaxed the requirements on spatial resolution but gained on the practical front, increasing data throughput and general adoptability of the approach, while still benefiting from most aspects of the shadow imaging approach. For instance, the relatively large FOV allowed us to monitor the movement of microglial processes over extended distances (>40 μm) and capture the macro-anatomical organization of acute brain slices, which would be impossible with a FOV of just 5 or 10 μm .

C. Some statements and conclusions made in the manuscript were not well supported by imaging experiments.

In the revised version, we have added more experiments in support of our statements and conclusions, in particular regarding the possibility to extract novel biological information / insights through our approach. Please see below for specific improvements in the revised manuscript.

D. Some important experiment details were missing.

We have made sure to not leave out any important experimental details in the revised version, expanding the explanations in the Figure legends and Methods sections. Also, please see below for the specific improvements we applied.

Please find my detailed questions and comments below:

1. For all imaging modalities applied in this work, their spatial resolutions were reported but all were indicated “data not shown”. In online Methods, the authors described how they assessed spatial resolution using sub-diffraction fluorescence beads. I would suggest the authors include their measurement results (i.e., measured point spread functions) as supplementary information.

We now provide the PSFs of all the imaging modalities in the supplementary information (**SI Fig. 1**), as requested.

2. The resolution measured from fluorescent beads does not represent the effective resolution supported by the “shadow imaging” method. Multiple factors would affect the practical resolution: extracellular labeling associated background and noise, sample induced aberrations, and inverted contrast processing. The effective resolution is an important parameter of “shadowing imaging” and should be carefully characterized. I would suggest the authors perform quantitative analysis, for example, show line profiles, or show the spectral representation of the images.

We now show line profiles across thin ECS spaces, e.g., between cell bodies, axons or perivascular spaces for all modalities, and report their full width at half maximum (FWHM) in the respective figures (**Fig. 1C**, **Fig. 3D**, **Fig. 5D**). The values are largely consistent with the PSFs recorded under ideal imaging conditions (fluorescent beads on coverslip) for the different modalities. However, they are, as expected, elevated due to the multiple adverse effects on image contrast and spatial resolution mentioned by the reviewer.

3. Line 101-102: the authors claimed that “it was possible...with little or no drop in signal-to-noise ratio (SNR) or sings of phototoxicity” – with “data not shown”. I would suggest the authors quantify image SNR and analyze how image SNR change during image acquisition.

We now support this statement by showing that indeed the SNR does not change over many image acquisitions (up to at least 100 frames). **Fig. 1E** shows a line profile across a thin ECS structure and its stability over time, where the peak of the signal (= the dye fluorescence from the ECS) remains unchanged, while the signal fluctuations coming from inside the adjoining cellular structures remain low over the entire acquisition time.

Also, how did the authors evaluate phototoxicity?

We evaluated phototoxicity by looking for cell blebbing and dye flooding into cells, which would be a sure sign of toxicity. The laser lesion experiments serve as a positive control, where processes and cell bodies can be seen blebbing and/or lysing, which we rarely, if ever, see under normal conditions, as illustrated in the video shown in the supplementary information.

Can the authors provide quantitative analysis of photo bleaching (e.g., signal change during image acquisition)?

We now show quantitative analysis of photo bleaching in the supplementary information (**SI Fig. 1C**) for COSHI and TUSHI, where this may be more of a concern than for light-sheet microscopy. The analysis clearly shows that there is no appreciable amount of photobleaching, as expected for the ECS labeling strategy. We measured a $13 \pm 12\%$ drop in intensity over 25 minutes of scanning (2 frames per minute) using TUSHI *in vivo*, which reflects the rate at which the dye gets cleared from the brain, corresponding to the $\sim 50\%$ drop in fluorescence we observed 4 hours after intraventricular dye injections (**Fig. 5C**), rather than photobleaching.

4. Fig 1F: I would suggest the authors show images from both channels as well as the merged, so that microglial morphology can be better visualized together with neurons. I am curious here that the video shows cell body shrinkage (the neuron on the left of the laser lesion) upon laser lesion – may the increased local fluorescence be autofluorescence from tissue debris? From morphology, I think the bright yellow signal came from microglia, but I am curious that why microglia in the lesion area displayed dark hollows inside the cell bodies? These hollows were not observed in Fig. 1E.

We thank the reviewer for the suggestion to show images from both channels and not just the merged channel, which illustrates the power of the combined positive/inverse labeling approach. The images are now shown in **SI Fig. 2A**.

The changes in local fluorescence reflect either changes in the shape and size of the ECS or dye uptake into the cells, i.e., damaged neurons or phagocytic microglia. We don't detect appreciable levels of auto-fluorescence when we induce laser lesions in the absence of dye in the ECS.

The 'dark hollows' inside the microglia reflect the uptake of fluorescent interstitial solution, which looks black after contrast inversion, representing one of the strengths of the shadow imaging approach, because it provides information on phagocytic and/or endocytic activity (or lysing, in the extreme) of all the cells in the field of view.

5. Line 42-43, the authors claimed that electron microscopy and magnetic resonance imaging "cannot be combined in situ with other powerful neuro-technologies, such as Ca²⁺ imaging, electrophysiology and optogenetics." This is not true, for example: (1) Lake et al., "Simultaneous cortex-wide fluorescence Ca²⁺ and whole-brain fMRI", *Nature Methods*, 2020, (2) Chen et al., "MRI-guided robotic arm drives optogenetic fMRI with concurrent Ca²⁺ recording", *Nature Communications*, 2019, and others.

We thank the reviewer for pointing out these studies. We have now corrected the claim about the incompatibility. Nevertheless, we think it remains challenging to combine EM or MRI with light microscopy, for the reason of having to transfer the sample between different instruments and for reason of scale, which are very different for fMRI and fluorescence imaging with subcellular resolution.

Here, we have demonstrated that it is possible to get unbiased and comprehensive anatomical data (something that only MRI and EM were thought to be good at) from widely available light microscopy modalities (i.e., confocal, 2P, or light-sheet microscopy), which lend themselves also to giving functional information with subcellular resolution, which is not possible with fMRI or cortex-wide Ca²⁺ imaging.

6. Line 119-120: point-scanning techniques... “often too slow for imaging Ca²⁺ transients and other dynamic biochemical activities”, which was not demonstrated with LISHI, either. The authors only stated that “in principle LISHI can be performed alongside high-speed imaging of ...” Real calcium imaging should be performed to make this conclusion. If the authors only demonstrated imaging with higher speed but not real functional imaging, I would suggest the authors tone down their conclusion here.

This echoes the comment by R1, and we have performed fast measurements of glutamate and calcium transients with the lattice light-sheet microscope. The proof-of-principle is now included in the manuscript (**Fig. 3F, G, H and I**) and as videos in the supplemental information (**SI Video 9A, B and 10A, B**).

7. For in vivo TUSHI, 8 μ L of dye was injected, which is a very large volume of injection. As a comparison, GCaMP virus transfection only requires 30~50 nL per injection site. I understand that the authors injected the dye not directly to the brain tissue but ipsilateral ventricle. But I wonder how invasive this procedure is and how practical this method is for longitudinal studies.

By optimizing the solubility of the dye and the detection sensitivity of the microscope, we have been able to reduce the volume to 3-5 μ l at slower injection rate, which is well tolerated by the animals. We have not tried to repeat the injections, but we are pursuing labeling approaches for repeated or long-term dye delivery (via a cannula or genetically encoded fluorescence in the cerebrospinal fluid). While these ideas are sound in principle, their realization is out of the scope of the study, but we now mention them in the paper.

8. Supposedly the dye only labels extracellular space and does not go into the cells. But in Fig. 2G, there are some blobs inside cell bodies showing contrast – why is that? These blobs were not observed in Figs 1, 3.

The blobs are dead or lysed cells filled with dye, which is expected for acute brain slices, where damaged cells on the surface are prone to take up the dye much more than in organotypic slices or in vivo, where most cells are intact. The shadow imaging approach can therefore serve as a way to assess cell viability of the slice preparation, which is known to depend strongly on animal health, slicing protocols, experimental skills and other unknown factors.

9. The home-built 2-photon microscope was equipped with an SLM to reduce aberrations when imaging deep inside brain tissues. The objective working distance is only 0.3 μ m. How deep did the authors image? Was aberration correction necessary in this case?

The imaging depth was up to 150 - 200 μ m, limited by the working distance of objective and the confocal detection mode (please also see our response to R1). The aberration correction proved useful for sharpening the PSF, correcting aberrations introduced by the microscope optics as well as the brain tissue, giving higher spatial resolution and image contrast, but the benefit is gradual, not all-or-none.

10. Line 107-108: the authors mentioned that “However, it remains unclear, which other cellular structures they come into contact with in the surrounding neuropil.” The readers would assume following experiments would answer this question; however, I did not see a conclusion drawn from COSHI imaging of microglia.

This echoes the comment by R1. We have included an exemplary analysis that demonstrates the feasibility to extract this anatomical information (**Fig. 2B**). However, a full and detailed analysis is out of the scope of this study. Instead, we have focused on microglial interactions with brain tissue in the pathological setting of a laser lesion.

11. Line 26-27: the authors stated that “current neuroanatomical imaging approaches either require tissue fixation, do not have cellular resolution or only give a fragmented view”. Did the authors want to say: “..., either do not have cellular resolution or only give a fragmented view”?

Our sentence is to convey that EM requires fixation, MRI doesn't have cellular resolution, while classical fluorescence microscopy only gives a fragmented view. The sentence suggested by the reviewer does not cover the case of EM, which offers a comprehensive view with ultrastructural resolution, but alas not in live tissue.

12. What are the depths of the two images shown in Fig. 1C?

We have replaced the images in **Fig. 1C** from another z stack. This stack was acquired at a depth of 25.5 μm with a z-step of 0.5 μm (31 steps total, taking us to depth of 10 μm at the end). The image shown was acquired at a depth of 14 μm . The full video is shown as **SI Video 1A**.

13. I believe Ref 15: (SPIE, San Francisco, France, 2019) should be California, USA.

Fixed, thanks.

Reviewer #3 (Remarks to the Author):

The authors extend a method of staining the extracellular space they introduced in 2018 for STED microscopy to diffraction-limited light microscopy in brain slices (confocal- and light-sheet microscopy) and in vivo (two-photon). As in their previous papers (Cell 2018, Glia 2021), they show that after contrast inversion, the shadow image can be combined with cell-specific fluorescent labels (of different color) to provide information about the position of the labeled cells in the tissue and neighboring structures. Using microglia and injected fluorescent tumor cells as examples, movements of these cells can be tracked by time-lapse shadow imaging. Their customized two-photon microscope is equipped with SLM wavefront correction, resulting in excellent visualization of minute details such as perivascular spaces and astrocytic endfeet. An interesting methodological detail is the injection of dye into the lateral ventricle, which provided sufficient staining of the extracellular space for several hours after cranial window implantation. The technique is therefore not suitable for repeated imaging over days or during behavior.

We thank the reviewer for the positive assessment of the quality of the images and our technical solution for labeling the ECS. We are working on labeling strategies for repeated imaging over days or even during behavior, but this is out of the scope of this study.

The manuscript provides convincing images and movies to highlight the possibilities of shadow imaging in vivo, but does not deliver any new biological insights.

We have worked on extracting more biological insights from shadow imaging, both in brain slices (please also see our response to Reviewer 1) and in vivo.

For the in vivo part, our images and analysis provide important information about the volume fraction of the ECS, which is sizable, confirming biophysical experiments (based on point-source diffusion measurements and modeling) and soundly rejecting the view based on EM analysis of perfusion-fixed brains, where the ECS appears essentially depleted (**Fig. 6C**).

In addition, we now present quantitative measurements on the size of perivascular spaces surrounding capillaries, which has never been done in the living brain to the best of our knowledge (**Fig. 5E, F**). In addition, these measurements disclose a very strong correlation between the width of the perivascular spaces and the diameter of the associated blood vessel (**Fig. 5G**). These sub-micron spaces could play an important role in brain fluid dynamics and be a part of a 'glymphatic system', where brain fluids are thought to pass from the perivascular space around arterioles into the ECS of the neuropil and is drained from there back into the perivascular space around venules. Shadow imaging in vivo thus makes it possible to study this elusive and likely important part of the neurovascular system, and investigate its role in supply and clearance mechanisms of the brain as well as vasoconstriction/dilation processes.

Interactions between microglia and implanted tumor cells would be very interesting to study, but the current manuscript stops short of combining the two application examples.

Indeed, it will be very interesting to study interactions between invading tumor cells and other brain cells like microglia in light of the emerging field of cancer neuroscience. Indeed, we are planning to use this approach in our future studies, but it is beyond the scope here.

On the positive side, methodological improvements are described in sufficient detail to make the paper a useful resource for labs that would like to implement the technique.

We thank the reviewer for this positive assessment.

REVIEWER COMMENTS

Reviewer #1 (Remarks to the Author):

I thank the authors for responding to the concerns raised in the previous review by performing additional experiments and presenting new data. In my opinion, however, the revised manuscript still falls short in shedding new scientific insights through the otherwise very elegant technique of shadow imaging. The results presented here, such as the motion of microglia processes, their phagocytic uptake, etc., are all very descriptive. It is not obvious how our understanding of microglia biology is advanced by these observations.

Similarly, it is not clear how the shadow imaging technique contributes to our understanding of the glutamate sensing and calcium signaling, as the dynamic information is all contained in the GluSnFR and the GCaMP6 signals. The authors stated that the shadow imaging provides the "anatomic context", but failed to demonstrate what insights are gained by visualizing the anatomic context.

My second major concern was about the volume fraction (VF) of the extracellular spaces. The authors added a new measurement using the normalized fluorescence intensity approach, but did not address the original concern that the binarized images using the two segmentation methods (Adaptive Thresholding and SpineJ) appeared so different from each other and also very different from the raw images.

The discrepancy among the three sets of data shown in Fig. 6C also calls into question the fidelity of the binarization methods. What is the rationale for presenting all three? Since the segmentation approach will get more accurate and reliable with improved spatial resolution, as stated in the rebuttal letter, and since the higher resolution method already exists (SUSHI), it seems odd that the authors would present the VF measurement using the inferior technique. In my opinion, it makes more sense in the current manuscript to focus on the normalized fluorescence intensity approach that does not depend on the spatial resolution (to the extent that the resolution is good enough to identify regions of pure ECS) and perform a more rigorous analysis with careful calibration of the detector response (to ensure the linearity of the intensity measurement).

Reviewer #2 (Remarks to the Author):

The authors have addressed my concerns and questions and have performed additional analyses and experiments. I think the revised manuscript is ready for publication.

Reviewer #3 (Remarks to the Author):

In the revised version, the authors added quantification to several application examples, demonstrating that the method can be used to address novel types of questions, e.g. the dynamic interaction of microglia with different neuronal compartments and the relationship between blood vessel diameter and perivascular volume. They also added a third method to estimate the ECS volume from their image stacks, resulting in much larger estimates (~33%). The reader is now left with estimates between 12 and 42%, which is not very useful. The authors mention that the normalization method could be compromised "by sub-linear signal detection" (very true!), but do not even test/show that their detection system operates far from saturation under the conditions of Fig.6A (e.g. by imaging a series of fluorescein dilutions and analyzing the pixel value histograms). To make a quantitative claim about ECS VF and then list reasons why the repeated measurements could be systematically wrong is not progress. I encourage the authors to choose the analysis method they consider most reliable, do the necessary controls (/corrections) for linearity, and provide the reader with their best estimate of ECS volume, not several numbers to choose from.

Response to reviewers' comments

We thank the reviewers for their additional critical comments and the opportunity to improve and correct our manuscript. The major issue concerned the estimation of the volume fraction of the extracellular space. We could fix the problem of the normalized fluorescence intensity approach by acquiring new images using lower laser powers to stay well within the linear range of the photodetector response.

The caveat of sub-linearity is a particularity of our setup, which was originally designed for 2-photon STED microscopy using single photon-counting avalanche photodiodes (APD), which have a very high sensitivity but a relatively narrow dynamic range compared to photomultiplier tubes (PMT) used in regular 2-photon microscopes.

Changes in the manuscript text are highlighted in yellow. Please find below our point-by-point answers.

Reviewer #1 (Remarks to the Author):

I thank the authors for responding to the concerns raised in the previous review by performing additional experiments and presenting new data. In my opinion, however, the revised manuscript still falls short in shedding new scientific insights through the otherwise very elegant technique of shadow imaging. The results presented here, such as the motion of microglia processes, their phagocytic uptake, etc., are all very descriptive. It is not obvious how our understanding of microglia biology is advanced by these observations.

We thank the reviewer for recognizing the elegance of the shadow imaging approach, but disagree with the assessment that the new results we have presented are too descriptive and don't provide new insights into microglial biology. Biological progress hinges on innovative imaging approaches to enable researchers to make more discriminating and complete observations of complex and dynamic live samples. Beyond just hinting at various lines of investigation as the reviewer rightfully criticized in the first round of review, we performed several specific analyses, all explained in detail in the manuscript. We believe we have made a strong case for the utility and interest of studying labeled microglia together with the shadow imaging technique. Here, we present findings only possible when combined with the unbiased labeling provided by shadow imaging, including the impact of neighboring cellular structures on microglial process velocity in response to a lesion and the subsequent triage of debris destined for phagocytosis. Likewise, we produced quantitative data on perivascular spaces, which are of great neurobiological interest.

Similarly, it is not clear how the shadow imaging technique contributes to our understanding of the glutamate sensing and calcium signaling, as the dynamic information is all contained in the GluSnFR and the GCaMP6 signals. The authors stated that the shadow imaging provides the "anatomic context", but failed to demonstrate what insights are gained by visualizing the anatomic context.

In a similar vein, we believe light-sheet shadow imaging together with functional imaging is a very promising application. In the original round of review, we were asked to demonstrate the feasibility of using light-sheet microscopy to carry out in tandem anatomical and fast functional imaging of biochemical activities, which we have now demonstrated for two different popular biosensors.

A variety of experimental model preparations and conditions (human brain organoids, brain development and plasticity, stroke and epilepsy) are likely to be accompanied by dynamic changes in anatomical landscape as well as cellular signaling. Major changes in ECS and astrocytic calcium signaling would be expected during brain edema (stroke) and seizure activity (epilepsy). Likewise, Alzheimer's disease, where enhanced glutamate signaling occurs due to reduced glutamate uptake, or LTP, where the retraction of astrocytic processes from synapses promotes glutamate spill over, are examples where the combination of glutamate imaging and anatomical context would be very interesting to apply. However, to pursue any of these lines of research in earnest clearly lies out of the scope of this study.

We anticipate that other researchers will follow up on this combination, providing new opportunities to study the complex interplay between functional signals and the anatomical structures and context that give rise to and shape them.

My second major concern was about the volume fraction (VF) of the extracellular spaces. The authors added a new measurement using the normalized fluorescence intensity approach, but did not address the original concern that the binarized images using the two segmentation methods (Adaptive Thresholding and SpineJ) appeared so different from each other and also very different from the raw images.

The discrepancy among the three sets of data shown in Fig. 6C also calls into question the fidelity of the binarization methods. What is the rationale for presenting all three? Since the segmentation approach will get more accurate and reliable with improved spatial resolution, as stated in the rebuttal letter, and since the higher resolution method already exists (SUSHI), it seems odd that the authors would present the VF measurement using the inferior technique. In my opinion, it makes more sense in the current manuscript to focus on the normalized fluorescence intensity approach that does not depend on the spatial resolution (to the extent that the resolution is good enough to identify regions of pure ECS) and perform a more rigorous analysis with careful calibration of the detector response (to ensure the linearity of the intensity measurement).

We thank the reviewer for this suggestion, which we have fully taken on board. We have removed the results from the binarization method, which should be reserved for super-resolution SUSHI data. We now include only data from the normalized fluorescence intensity approach, which is independent of the spatial resolution of the microscope.

Addressing concerns about systematic errors for this approach, we have acquired new images and carried out calibration experiments to determine the linearity of the detector response.

Firstly, to identify regions of pure ECS, we looked for areas that included blood vessels with large perivascular spaces, i.e., large enough to contain the 2-photon excitation spot, and thus provide a faithful measure of pure ECS.

Secondly, we ensured the linearity of the fluorescence intensity measurements by adjusting the 2-photon laser power for a given imaging depth and labeling intensity to keep photon count rates within the linear range of the single-photon counting APD, whose maximal count rate is nominally 37 Mcps ('mega counts per second').

To determine the linear range of our APD, we imaged a series of Calcein dilutions (see below, Fig. 1). The detector response scaled with the dye concentration in a highly linear way for photon count rates of up to 4 Mcps ($r^2 = 0.9987$).

For the new VF analysis, we acquired images *in vivo* where the mean count rates in the neuropil were between 0.2 and 1 Mcps and the maximal count rates in the pure ECS regions between 1 and 4 Mcps, i.e., well within the linear range of the detector response (Fig. 2). Staying within this limit, we estimated the VF values *in vivo* to be on average 23% (23 ± 0.7 %, mean \pm SEM; N = 4 mice, n = 36 regions of interest; Fig. 6C). By comparison, in our previous measurements, the mean count rates in the neuropil were between 1.8 and 3.4 Mcps, which was in the linear range, but the maximal count rates (pure ECS) were between 5 and 16 Mcps, i.e., outside of the linear detector response, leading to an overestimate of the VF, because the sub-linearity gave a falsely low reading for the pure ECS reference signal. In summary, we believe that these new VF estimates are trustworthy because we made sure to stay within the linear range of the detector response and selected regions of pure ECS that were large enough for the resolution of our microscope.

Reviewer #2 (Remarks to the Author):

The authors have addressed my concerns and questions and have performed additional analyses and experiments. I think the revised manuscript is ready for publication.

Thank you for your approval!

Reviewer #3 (Remarks to the Author):

In the revised version, the authors added quantification to several application examples, demonstrating that the method can be used to address novel types of questions, e.g. the dynamic interaction of microglia with different neuronal compartments and the relationship between blood vessel diameter and perivascular volume. They also added a third method to estimate the ECS volume from their image stacks, resulting in much larger estimates (~33%). The reader is now left with estimates between 12 and 42%, which is not very useful. The authors mention that the normalization method could be compromised "by sub-linear signal detection" (very true!), but do not even test/show that their detection system operates far from saturation under the conditions of Fig.6A (e.g. by imaging a series of fluorescein dilutions and analyzing the pixel value histograms). To make a quantitative claim about ECS VF and then list reasons why the repeated measurements could be systematically wrong is not progress. I encourage the authors to choose the analysis method they consider most reliable, do the necessary controls (/corrections) for linearity, and provide the reader with their best estimate of ECS volume, not several numbers to choose from.

We appreciate the reviewer's valuable suggestion, which prompted us to make several important adjustments and corrections.

Specifically, we have removed the analyses based on the binarization method, as it should only be applied to super-resolution SUSHI images, where the ECS spatial structure can be much better resolved.

In the revised manuscript, we now only include results based on the normalized fluorescence intensity approach, which is inherently independent of the microscope's spatial resolution.

As suggested by the reviewer, we have determined the linear range of the detector response by acquiring images of a dilution series with the fluorescent dye Calcein (see below, Fig. 1). Fitting a linear function to the data shows that for photon count rates of up to at least 4 Mcps (Mega counts per second) the detector response scales with the dye concentration in a highly linear way (goodness of fit, $r^2 = 0.9987$), indicating that as long as the count rates stay below this value, the normalized fluorescence approach should yield faithful VF estimates.

As our previous VF analysis had included data for pure ECS that were outside of the linear range of the detector response, we acquired new images with lower 2-photon laser powers to stay within the linear range. As a result, we obtained lower VF estimates ($23 \pm 0.7\%$, $N = 4$ mice, $n = 36$ regions of interest; Fig. 6C), which should be accurate now.

Figure 1: Photon count rate plotted against Calcein concentration *in vitro* to determine the linear range of detector response (excited by 15 mW at 920 nm 2-photon laser power with a 20 μ s pixel dwell time; $N = 6$). Up to at least 4 Mcps, the detector response is highly linear.

Figure 2: Ranges of photon count rates in neuropil (*left*) and photon count rates in pure ECS regions (*right*) used for ECS VF analysis. The images were acquired *in vivo* with variable 2-photon laser power depending on imaging depth but with a fixed pixel dwell time of 200 μ s. Both ranges are well within the linear operating range of the APD detector (cf. Fig. 1), thus far away from saturation (which nominally occurs at 37 Mcps).

REVIEWERS' COMMENTS

Reviewer #1 (Remarks to the Author):

Please see my comments to the editors.

Reviewer #3 (Remarks to the Author):

In their revision, the authors have addressed the methodological concerns about detector saturation and arrive at a credible estimate of extracellular volume in vivo, which is surprisingly different from EM estimates after tissue fixation. I have no further comments or concerns.